# Hard magnetic properties in nanoflake van der Waals Fe$_3$GeTe$_2$

Cheng Tan[1], Jinhwan Lee[2], Soon-Gil Jung [3], Tuson Park[3], Sultan Albarakati[1], James Partridge[1], Matthew R. Field[1], Dougal G. McCulloch[1], Lan Wang[1] & Changgu Lee[4]

Two-dimensional van der Waals materials have demonstrated fascinating optical and electrical characteristics. However, reports on magnetic properties and spintronic applications of van der Waals materials are scarce by comparison. Here, we report anomalous Hall effect measurements on single crystalline metallic Fe$_3$GeTe$_2$ nanoflakes with different thicknesses. These nanoflakes exhibit a single hard magnetic phase with a near square-shaped magnetic loop, large coercivity (up to 550 mT at 2 K), a Curie temperature near 200 K and strong perpendicular magnetic anisotropy. Using criticality analysis, the coupling length between van der Waals atomic layers in Fe$_3$GeTe$_2$ is estimated to be ~5 van der Waals layers. Furthermore, the hard magnetic behaviour of Fe$_3$GeTe$_2$ can be well described by a proposed model. The magnetic properties of Fe$_3$GeTe$_2$ highlight its potential for integration into van der Waals magnetic heterostructures, paving the way for spintronic research and applications based on these devices.

[1] School of Science, RMIT University, Melbourne VIC 3001, Australia. [2] School of Mechanical Engineering, Sungkyunkwan University, Suwon 440-746, Republic of Korea. [3] Center for Quantum Materials and Superconductivity (CQMS) and Department of Physics, Sungkyunkwan University, Suwon 440-746, Republic of Korea. [4] School of Mechanical Engineering and SKKU Advanced Institute of Nanotechnology (SAINT), Sungkyunkwan University, Suwon 440-746, Republic of Korea. These authors contributed equally: Cheng Tan, Jinhwan Lee. Correspondence and requests for materials should be addressed to L.W. (email: lan.wang@rmit.edu.au) or to C.L. (email: peterlee@skku.edu.kr)

wo-dimensional (2D) van der Waals (vdW) materials have received considerable attention since the successful isolation of graphene[1, 2]. Studies on these materials have revealed novel optical[3, 4] and electronic[5, 6] properties. Moreover, employing heterostructures based on these 2D vdW materials has revealed further interesting properties and suggested applications[7–9]. vdW magnets were known more than 50 years ago[10–12], but interest has been renewed with the emergence of 2D materials. In the last few years, Raman spectroscopy[13–17] and electron transport measurements[18, 19] have been performed on 2D magnets. Importantly, 2D ferromagnetism has been discovered very recently in two insulating vdW materials, $Cr_2Ge_2Te_6$[20] and $CrI_3$[21] and novel devices based on vdW ferromagnetic heterostructures have been demonstrated[18, 22, 23]. The opportunity exists to design and fabricate many devices based on vdW magnets. For example, vdW magnetic insulator can magnetize 2D topological insulators by the magnetic proximity effect and thereby generate the quantum anomalous Hall effect in these materials[24–27]. vdW ferromagnetic metals can be employed in spin–orbit torque devices when stacked with vdW metals with strong spin–orbit interactions[28–33]. However, in order to exploit ferromagnetic vdW materials as building blocks for vdW heterostructure-based spintronics, a ferromagnetic vdW metal with a hard magnetic phase and a large magnetic remanence to saturated magnetization ($M_R/M_S$) ratio is essential. This kind of vdW ferromagnetic metal is scarce.

Among all the predicted and experimentally observed vdW ferromagnetic materials[19–21, 23, 34–43], a very promising ferromagnetic metal is $Fe_3GeTe_2$ (FGT), which exhibits a Curie temperature ($T_C$) near 220 K in its bulk state[42]. Previous experimental work has shown that bulk single crystalline FGT has a ferromagnetic state with a very small $M_R/M_S$ ratio and coercivity at all temperatures[42–45], suggesting limited potential as a building block for vdW magnetic heterostructures. However, the $M_R/M_S$ ratio and coercivity of a magnetic material strongly depend on its domain structure[46–48], which is thickness dependent. Furthermore, recent research[49] shows that molecular beam epitaxy (MBE)-grown wafer-scale FGT thin films have improved magnetic properties. These findings motivate us to investigate the magnetic properties of exfoliated FGT nanoflakes of various thicknesses using anomalous Hall effect measurements.

Here, we report anomalous Hall effect measurements on single crystalline FGT nanoflakes and show that their magnetic properties are highly dependent on thickness. Importantly, by reducing the thickness to less than 200 nm, a hard magnetic phase with large coercivity and near square-shaped hysteresis loop occurs. These characteristics are accompanied by strong perpendicular magnetic anisotropy, making vdW FGT a ferromagnetic metal suitable for vdW heterostructure-based spintronics. By employing criticality analysis, the existence of magnetic coupling with a coupling length of ~5 vdW layers between vdW atomic layers is estimated in FGT. Finally, we propose a model to describe the hard magnetic behaviour of FGT thin flakes. This model is suitable for other vdW ferromagnetic thin films or nanoflakes with strong perpendicular anisotropy and square-shaped magnetic loops.

## Results

**Thickness-dependent anomalous Hall effect**. In a ferromagnetic material, the relationship between the Hall resistance and the

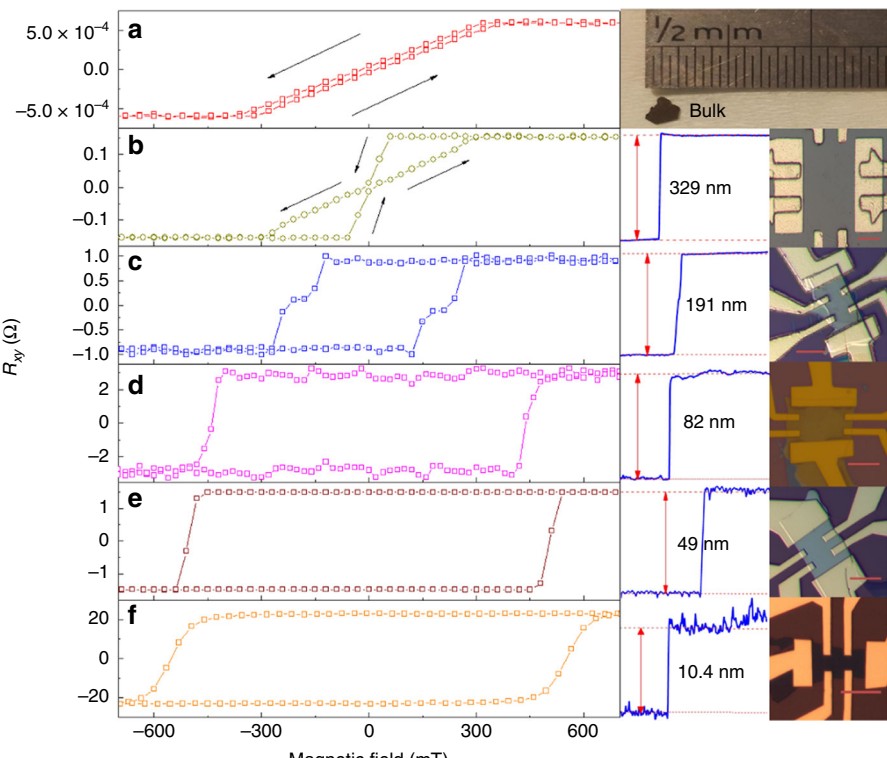

**Fig. 1** $R_{xy}(B)$ for FGT nanoflakes of various thicknesses at 2 K. Each red scale bars represents 10 μm. **a** Bulk device, $L \times W \times T = 1.8$ mm × 0.8 mm × 0.3 mm. $M_R/M_S = 0.0715$. **b** A device with a thickness of 329 nm, $L \times W \times T = 44.4$ μm × 49.4 μm × 329 nm. $M_R/M_S = 0.0807$. **c** A device with a thickness of 191 nm, $L \times W \times T = 14.7$ μm × 9.25 μm × 191 nm. $M_R/M_S = 0.9757$. **d** A device with a thickness of 82 nm, $L \times W \times T = 10.3$ μm × 19.5 μm × 82 nm. $M_R/M_S = 0.9839$. **e** A device with a thickness of 49 nm, $L \times W \times T = 12.6$ μm × 13.1 μm × 49 nm. $M_R/M_S = 0.9980$. **f** A device with a thickness of 10.4 nm, $L \times W \times T = 12.7$ μm × 8.79 μm × 10.4 nm. $M_R/M_S = 0.9973$

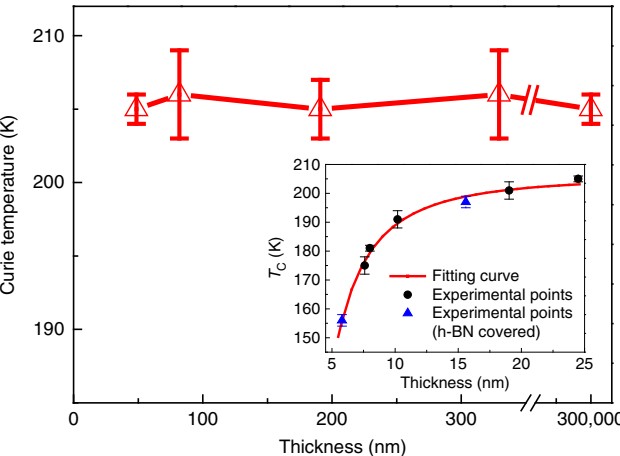

**Fig. 2** Thickness dependence of the Curie temperature ($T_C$). Excepting the device with a thickness of 329 nm, the $T_C$s of the devices were determined by the temperature point when the remanence becomes zero. The error bar is determined by the noise level around the point when the remanence becomes zero (Supplementary Fig. 8). The $T_C$ of the device with a thickness of 329 nm was determined from its $\rho_{xx}(T)$ curve. Inset shows the thickness dependence of the $T_C$s in nanoflakes with thicknesses from 5.8 to 25 nm with fitting curve. Blue dots are the $T_C$s for the two ultra-clean devices that were covered by h-BN and PMMA in a glove box ($O_2$ <0.1 p.p.m. and $H_2O$ <0.1 p.p.m.)

applied magnetic field is given by Eq. (1).

$$R_{xy} = R_0 B_Z + R_S M_Z \qquad (1)$$

here $R_{xy}$ is the Hall resistance, which is composed of a normal Hall resistance (the first term in Eq. (1)) and an anomalous Hall resistance (the second term in Eq. (1)). $B_Z$ and $M_Z$ are the applied magnetic field and the sample magnetic moment perpendicular to the sample surface, respectively. The anomalous Hall resistance is proportional to $M_Z$. As FGT is a metallic ferromagnetic material, the normal Hall resistance is negligible compared with the anomalous Hall resistance in the magnetic field range of interest. Hence, the shape of the $R_{xy}$ vs $B$ loop is actually the same as that of the $M_Z$ vs $B$ loop. The coercivity and $M_R/M_S$ ratio can be obtained from the $R_{xy}(B)$ curve.

We measured the longitudinal resistance $R_{xx}$ and transversal resistance $R_{xy}$ of 11 FGT nanoflake devices with thickness from 5.8 to 329 nm and a bulk FGT crystal. The $R_{xy}(B)$ of selected FGT nanoflake devices at 2 K are shown in Fig. 1b–f with the $R_{xy}(B)$ of the bulk FGT crystal (Fig. 1a). The applied magnetic field was perpendicular to the surfaces of the samples. The coercivity and $M_R/M_S$ ratio of the bulk crystal sample at 2 K are only 21.6 mT and 0.07, respectively. These characteristics agree well with those measured using a magnetometer[42]. However, the exfoliated nanoflakes displayed different magnetic properties. The nanoflakes with thicknesses of 329 and 191 nm displayed magnetic loops resembling two magnetic phases with different coercivities. As shown in Fig. 1b, c, when the magnetic field sweeps from the positive saturation field to the negative saturation field, the $R_{xy}$ values decrease sharply at a lower negative magnetic field and then decrease again at a higher negative field, which is similar to the behaviour of the coexistence of two phases. When the thicknesses of the FGT nanoflakes decrease further, the 'two-phase' behaviour disappears. As shown in Fig. 1d–f, the $R_{xy}(B)$ loops of the three samples (with thicknesses of 82, 49 and 10.4 nm, respectively) display a near square shape, indicative of a single hard magnetic phase. The coercivities of these three samples are much larger than

those of the samples with 329 and 191 nm thickness and exceed 400 mT at 2 K. As FGT gradually evolves from a soft phase (bulk) to a single hard phase (82, 49 and 10.4 nm), we speculate that the 'two-phase' behaviour in the nanoflakes with thicknesses of 329 and 191 nm is due to the gradual evolution of the domain structure. The $M_R/M_S$ ratios of the 191, 82, 49 and 10.4 nm thickness nanoflakes are near 1 at 2 K, demonstrating that all their magnetic moments remain aligned perpendicular to the sample surfaces at the remanence point. The magnetic moments flip abruptly to the opposite direction at the coercive field. In the magnetic field regime away from the coercivity, the four nanoflakes behave like a single magnetic domain with a strong perpendicular anisotropy. The magnetic domains only appear and flip to the opposite direction near the coercivity. As bulk single crystalline FGT also shows a strong perpendicular anisotropy, the anisotropy should be induced by the crystalline field. With increasing temperature, FGT gradually evolves from ferromagnetic to paramagnetic state.

**$T_C$ and interlayer magnetic coupling.** The $T_C$ of each FGT nanoflake device can be determined from the temperature point where the remanence value goes to zero[20], as in the Supplementary Fig. 8. The dependence of $T_C$ on thickness (from 0.3 to 49 nm) is shown in Fig. 2, from which we conclude that the $T_C$ of the FGT nanoflakes is almost independent of thickness in this range. As shown in the inset of Fig. 2, the $T_C$ decreases sharply as the thickness decreases from 25 to 5.8 nm. This behaviour is similar to that in $Cr_2Ge_2Te_6$[20], but is different from the behaviour in $CrI_3$[21]. It should be emphasized that nine of the 11 devices were fabricated in ambient condition with an air exposure of ~7 min. The other two ultra-clean devices were made in a glove box ($O_2$ <0.1 p.p.m., $H_2O$ <0.1 p.p.m.) with h-BN and poly(methyl methacrylate) (PMMA) covering (details in methods and Supplementary Note 6). The two batches of devices show the same magnetic characteristics (Fig. 1d–f and Supplementary Fig. 10d). The theory of critical behaviour[50] reveals that the finite thickness of flakes limits the divergence of the spin–spin correlation length at the $T_C$. As FGT is an itinerant metallic system, its spin–spin coupling should extend for many atomic layers, even along the out of plane directions. The spin–spin coupling range along the out of plane direction can be fitted as shown in the inset of Fig. 2 using the formula[51]

$$1 - T_C(n)/T_C(\infty) = [(N_0 + 1)/2n]^\lambda, \qquad (2)$$

where $T_C$ is the Curie temperature, $n$ is the number of atomic layers of a flake, $N_0$ is the spin–spin coupling range, and $\lambda$ is a universal critical exponent. A best fitting to the data requires $\lambda = 1.66 \pm 0.18$ and $N_0 = 4.96 \pm 0.72$ monolayers. The fitted $\lambda = 1.66$ is near the value of a standard three-dimensional (3D) Heisenberg ferromagnetism[52]. The correspondence achieved using a single fitting curve also indicates that FGT nanoflakes with a thickness of more than 5 vdW layers are still 3D ferromagnets. If FGT nanoflakes evolve from 3D ferromagnetism to 2D ferromagnetism from 25 to 5.8 nm, we should be able to obtain two fitting curves with different critical exponents. However, the data does not show this behaviour, which further confirms 3D magnetism when FGT thickness >5.8 nm. Scaling behaviour[40, 41] near the $T_C$s of samples with thicknesses from monolayer to >10 nm should reveal the evolution of the magnetism from 3D to 2D with decreasing thickness in FGT. As the focus of this paper is revealing the hard magnetic properties of FGT nanoflakes and their suitability for future spintronic applications, we propose this scaling analysis as future work.

**Detailed measurements and modified Stoner–Wohlfarth model.** More detailed measurements were performed on the 10.4

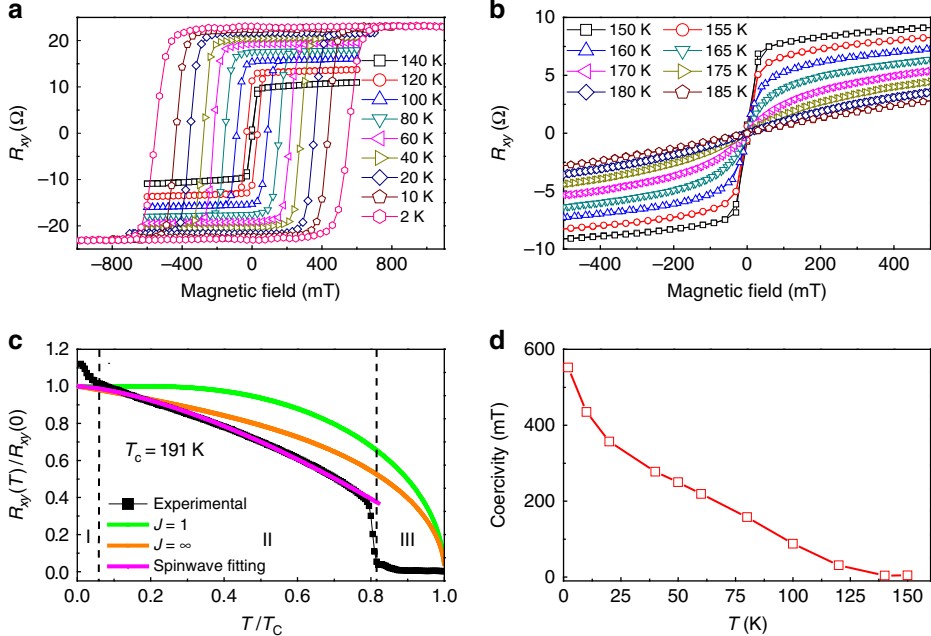

**Fig. 3** Anomalous Hall effect measurements performed on 10.4 nm thickness FGT device. **a** $R_{xy}(B)$ loops in the temperature regime from 2 to 140 K in which the FGT nanoflake shows ferromagnetic properties. **b** $R_{xy}(B)$ curves in the temperature regime from 150 to 185 K. The FGT nanoflake shows zero coercivity and remanence when $T > 150$ K. **c** Normalized $R_{xy}(T)$ curve and three fitting curves based on the mean field theory $J = 1$, $J = \infty$, and the spin wave theory, respectively. Three magnetic regimes exist from 2 K to $T_C = 191$ K. Regime I (2-10 K) is an unknown phase requiring further investigation. Regime II (10-155 K) is a hard ferromagnetic phase. Regime III (155-191 K) is a phase with zero coercivity and remanence. **d** The temperature dependence of coercivity from 2 to 150 K

nm thickness nanoflake. In Figs. 3a and 3b, the $R_{xy}(B)$ loops from this sample, measured with perpendicular applied magnetic field, are plotted at various temperatures. There is a clear evolution with increasing temperature. At 2 K, the $R_{xy}(B)$ loop is nearly square-shaped with a large coercivity of 552.1 mT and $M_R/M_S \sim 1$, revealing alignment of spins due to strong perpendicular anisotropy. The $R_{xy}(B)$ loops remain approximately square up to 155 K. Figure 3d displays the temperature dependence of coercivity in this temperature regime. When the temperature exceeds $T_C$ (~191 K), the nanoflake becomes paramagnetic.

We also measured the temperature dependence of the $R_{xy}$ at the remanence point of the sample (the measurement and data process details of $R_{xy}(T)$ are shown in the methods). The $R_{xy}(T)$ of the 10.4 nm nanoflake is shown in Fig. 3c. $R_{xy}(0)$ is an extrapolation to $T = 0$ K from region II based on the spin wave model as discussed later. Results from other samples are shown in the Supplementary Note 2. Below 155 K, the FGT nanoflake exhibits a ferromagnetic phase with near square-shaped magnetic loop. The abrupt decrease of the magnetic moment around 155 K in Fig. 3c indicates a first order magnetic phase transition not yet known, but likely related to the competition between the perpendicular anisotropic energy and the thermal agitation energy. As the bulk FGT single crystal also shows a phase transition with gradually changed magnetic moment around 155 K[45], we speculate that the sharper phase transition in the FGT nanoflake is due to its decreased thickness, which induces single domain behaviour at the remanence. In the temperature regime from 155 K to $T_C$, the FGT nanoflake displays a ferromagnetic phase with very small coercivity and remanence. The $R_{xy}(T)$ reveals another phase transition near 10 K, where the remanence increases sharply with decreasing temperature, indicative of the formation of new spins and magnetic moments. Further understanding of the phases present in this temperature regime would require neutron scattering measurements, which are beyond the scope of this article. Figure 3c also shows the temperature

dependence of $R_{xy}(T)$ fitted from 2 to 150 K using the mean field theory (the Brillouin function) and the spin wave theory (details in Supplementary Note 5). The $R_{xy}(T)$ behaviour of the 10.4 nm thickness sample cannot be fitted using mean field theory, but agreement with the spin wave theory for a 3D ferromagnet is good. This provides further evidence that an FGT nanoflake remains a 3D magnetic system when its thickness exceeds five layers. Our experimental results contain information required to construct a correct model for FGT. First, as shown in $R_{xy}(B)$ measurements (Figs. 1, 3 and 4), FGT has a very strong perpendicular anisotropy due to the crystalline field. Second, the thickness dependence of $T_C$ as shown in Fig. 2 indicates the existence of magnetic coupling between atomic layers in FGT with an estimated coupling range of about 5 vdW layers. Therefore, a correct Hamiltonian describing FGT should include a perpendicular anisotropic energy, an in-plane spin–spin interaction energy, an out of plane spin–spin interaction energy and a Zeeman energy induced by the applied magnetic field.

The evolution of $R_{xy}$ hysteresis loops with the angle $\theta$ between the applied magnetic field and the direction perpendicular to the sample surface (i.e. the direction of magnetic anisotropy) for the 10.4 nm flake at 2 K is shown in Fig. 4a, b. As mentioned prior, the spins in the FGT nanoflakes align to one direction due to the strong perpendicular anisotropy when the temperature is below 155 K. Magnetic domains only appear near the coercive field. This simple magnetic structure makes it possible to construct a model to describe the behaviour of the coercivity. When a magnetic field is applied to a single domain ferromagnetic material with uniaxial anisotropy, the energy of the magnetic system is composed of the magnetic anisotropic energy and the Zeeman energy,

$$E(T) = K_A(T)V_S \sin^2(\phi - \theta) - M_S(T)BV_S \cos\phi, \quad (3)$$

where $K_A$ is the magnetic anisotropic energy per volume, $V_S$ is the volume of the sample, $\phi$ is the angle between the magnetic field

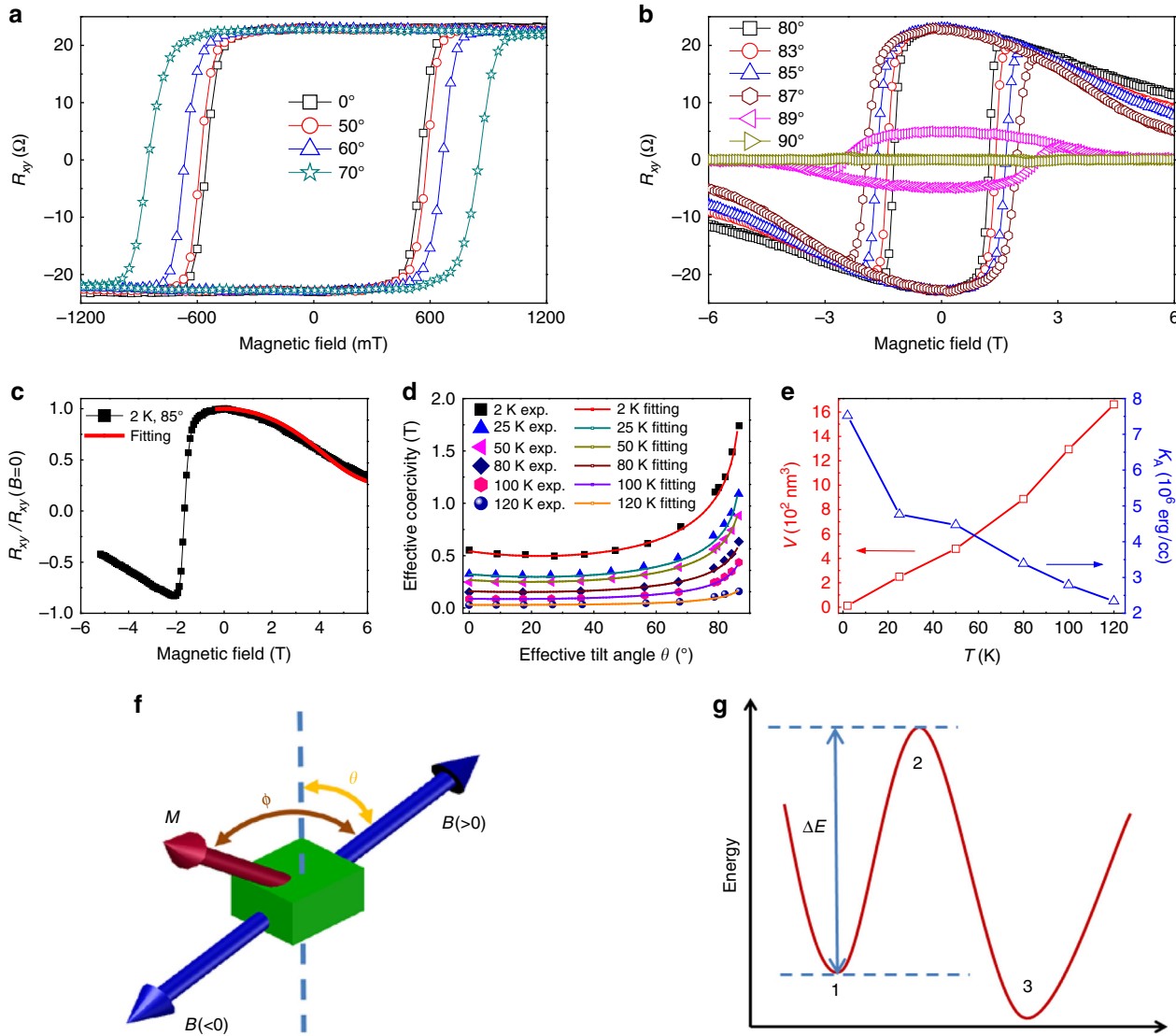

**Fig. 4** Angular dependent Hall effect measurements and modified Stoner–Wohlfarth model. **a, b** $R_{xy}(B)$ loops at different angles between the applied magnetic field and the direction perpendicular to the surface of the nanoflake with a thickness of 10.4 nm. At 0°, the surface of the nanoflake is perpendicular to the magnetic field. **c** Normalized $R_{xy}(B)$ curve at 2 K measured at 85° from 6 to −6 T. The fitting curve is based on the Stoner–Wohlfarth model. **d** The effective angular dependence of effective coercivities at different temperatures. The solid curves are the fitting curves based on a modified Stoner–Wohlfarth model. From **b** we can tell that the remanences of the $R_{xy}$ loops at angles >85° show pronounced decreases and a divergent coercivity value was obtained at 90°, an angle beyond the range included by the modified Stoner–Wohlfarth model. Based on this experimental result, the theoretical fittings were only performed to 85°. **e** The temperature dependence of $K_A$ and $V$. $K_A$ was fitted by the Stoner–Wohlfarth model, while $V$ was fitted based on the fitted $K_A$ and the modified Stoner–Wohlfarth model (Supplementary Note 4). **f** Illustration of the variables used in the Stoner–Wohlfarth model. The dashed line is the easy axis of the magnetic anisotropy in FGT nanoflakes. **g** Schematic diagram of a magnetic system changing from a metastable to an unstable state

and the magnetic moment, and $\theta$ is the angle between the magnetic field and the direction of magnetic anisotropy (Fig. 4f), $M_S(T)$ is the magnetic moment of a unit volume FGT at temperature $T$, and $B$ is the applied magnetic field. With an applied magnetic field $B$ and the angle $\theta$, we can calculate $\phi$ from Eq. (4), the well-known Stoner–Wohlfarth model[53]

$$\frac{\partial E(T)}{\partial \phi} = 2K_A(T)V_S \sin(\phi - \theta)\cos(\phi - \theta) + M_S(T)BV_S\sin\phi = 0,$$

(4)

For thin FGT nanoflakes with perpendicular anisotropy, the demagnetization effect[46] should be considered. Consequently, the

applied magnetic field $B$ and angle $\theta$ in Eqs. (3) and (4) should be modified to the effective magnetic $B_{eff}$ and angle $\theta_{eff}$. Further detail is shown in Supplementary Note 4. Using this model, we fitted the $R_{xy}(B)$ curve with $\theta = 85°$ to obtain the unit magnetic anisotropic energy $K_A$ at 2, 25, 50, 80, 100, and 120 K, as shown in Fig. 4e (additional details provided in Supplementary Note 4). Figure 4c shows the fitting curve for the 2 K data. It should be emphasized here that all the $R_{xy}(B)$ with various $\theta$ values at different temperatures in the magnetic regime away from the coercivity can be well fitted by the Stoner–Wohlfarth model. The reason for using $\theta = 85°$ loops for the $K_A$ fitting is that a magnetic loop of small $\theta$ value is nearly a straight line without curvature in the magnetic regime away from the coercivity, which is not suitable for obtaining a reliable $K_A$.

As magnetic domains appear in the magnetic field regime near coercivity, to describe the angular dependence of coercivity, the flip of magnetic domains near coercivity should be included in model. By considering the thermal agitation energy and utilizing the fitted $K_A$ values, a modified Stoner–Wohlfarth model (details in Supplementary Note 4) can be used to describe the angular dependence of coercivity. As shown in Fig. 4g, if the system can be thermally excited from a metastable state (state 1) to an unstable state (state 2), the magnetic moment can then be flipped to a final stable state (state 3). The energy difference between states 1 and 2 is $\Delta E$. With increasing magnetic field $B$ (more negative $B$), the energy difference $\Delta E$ between the metastable state 1 and the unstable state 2 decreases. In the modified Stoner–Wohlfarth model, we make two assumptions:

1. At a certain $B$ field, the thermal agitation energy is large enough to overcome the $\Delta E$ in a standard experimental time (100 s) causing the magnetic moment to flip. As the FGT shows a nearly square-shaped $R_{xy}$ loop (magnetization loop), we can assume that this $B$ field is the coercive field, which is a reasonable approximation due to the sharp transition of the magnetic moments.

2. When the first domain flips under an applied magnetic field, other un-flipped magnetic moments will generate an effective field on the magnetic moment in the first flipped domain. The processes of domain flip, expansion, and interaction are complex. Micro-magnetic simulation is required to provide a detailed description, which is beyond the scope of this paper. Here, a parameter $a(T)$ is used to describe the mean field interaction between the flipped and un-flipped magnetic moments.

Based on the proposal of Neel and Brown[54, 55], we use $\Delta E = 25k_BT$ as the barrier height where the magnetic moment starts to flip (details in Supplementary Note 4). We thus obtain

$$\begin{aligned}&\left[K_A(T)V(T)\sin^2(\phi_2-\theta)-[1-a(T)]V(T)M_S(T)B\cos\phi_2\right]\\&-\left[K_A(T)V(T)\sin^2(\phi_1-\theta)-[1-a(T)]V(T)M_S(T)B\cos\phi_1\right]=25k_BT,\end{aligned}$$
$$(5)$$

where $V(T)$ is the volume of the first flipped domain at $T$, $a(T)$ is the parameter describing the effective field due to the coupling between the first flipped domain and the un-flipped magnetic moments at temperature $T$, which affects the Zeeman energy. The value of $a(T)$ lies between 0 and 1. $\phi_1$ and $\phi_2$ are the angles between the applied magnetic field and the magnetic moment for states 1 and 2, respectively. These angles are calculated using Eq. (4). Due to the demagnetization effect[46], $B$ and $\theta$ here should also be modified to $B_{eff}$ and $\theta_{eff}$ (Supplementary Note 4).

Using $V(T)$ and $a(T)$ as fitting parameters in Eq. (5) in conjunction with the modified Stoner–Wohlfarth model provides excellent agreement with the experimental angular dependence of coercivity at various temperatures (Fig. 4d), which further confirms the single domain behaviour induced by the strong perpendicular anisotropic energy in FGT in the field regime away from coercivity. The volume of the first flipped magnetic domain near the coercivity $V(T)$ is important for understanding the magnetic dynamics of a ferromagnetic materials. This volume $V$ and the perpendicular anisotropic energy $K_A$ at different temperatures are shown in Fig. 4e. The modified Stoner–Wohlfarth model proposed here is suitable for describing the magnetic behaviour of 2D vdW ferromagnetic materials with strong perpendicular anisotropy and near square-shaped loop.

To conclude, FGT nanoflakes are vdW 2D metallic ferromagnets with large coercivity, $M_R/M_S$ ratio of 1, relatively high $T_C$ and strong perpendicular anisotropy. Exploitation of this material in various vdW magnetic heterostructures with properties, such as giant magnetoresistance and tunnelling magnetoresistance, as well as vdW spin–orbit torque heterostructures is expected to

yield exciting results. This discovery paves the way for a new research field, namely, vdW heterostructure-based spintronics.

## Methods

**Single crystal growth**. Single crystal FGT was grown by the chemical vapor transport method. High-purity Fe, Ge and Te were blended in powder form with molar proportions of 3:1:5 (Fe:Ge:Te). Iodine (5 mg/cm$^2$) was added as a transport agent and the mixed constituents were sealed into an evacuated quartz glass ampoule. This ampoule was placed in a tubular furnace, which has a temperature gradient between 700 and 650 °C. The furnace center temperature was ramped up to 700 °C with a heating rate of 1 °C per minute and was maintained at the set point for 96 h. To improve crystallinity, the ampoule was slowly cooled down to 450 °C for over 250 h. Below 450 °C, the furnace was cooled more rapidly to room temperature.

**Device fabrication and measurement**. First, the single crystalline FGT was mechanically exfoliated and placed on a Si substrate with a 280 nm thickness SiO$_2$ layer. Then, Cr/Au (5 nm/100 nm) contacts were patterned by photolithography and e-beam lithography. During intervals between processing, the sample was covered by a polydimethylsiloxane (PDMS) film and stored in an evacuated glass tube (~$10^{-6}$ Torr). The sample was exposed to ambient for no more than 7 min throughout the fabrication procedure. For the h-BN covered devices, 5 nm thick Pt contacts were firstly fabricated on Si/SiO$_x$ substrate in ambient condition. In a glove box (O$_2$, H$_2$O <0.1 p.p.m.), an exfoliated FGT flake was then dry transferred by PDMS onto the contacts to form very good ohmic contacts. Thereafter, a large h-BN flake was dry transferred to cover the FGT flake to prevent oxidations in measurements. Finally, the sample was covered by PMMA to prevent any possible oxidization. The transport measurements were performed in a Physical Property Measurement System (Ever Cool II, Quantum Design, San Diego, CA, USA) with 9 T magnetic field.

**Hall effect measurement and data processing**. Because of the non-symmetry in our nanoflake devices, the measured Hall resistance was mixed with the longitudinal magnetoresistance. We processed the data by using $(R_{xyA}(+B)-R_{xyB}(-B))/2$ to eliminate the contribution from the longitudinal magnetoresistance, where $R_{xyA}$ is the half-loop sweeping from the positive field to the negative field, $R_{xyB}$ is the half-loop sweeping from the negative field to the positive field, and $B$ is the applied magnetic field. We also measured the $R_{xy}(T)$ at the remanence point for most of the samples. To measure the $R_{xy}(T)$ at the remanence point, the magnetic moment of samples was first saturated by a 1 T magnetic field and then the magnetic field was decreased to zero (the remanence point). Finally, the temperature dependence of the $R_{xy}$ at remanence was measured when the temperature was increased from 2 to 300 K. In order to eliminate the non-symmetry effect of the device, we measured $R_{xy}$ (remanence) with both 1 and $-1$ T saturation. The real $R_{xy}(T)$ at remanence without $R_{xx}$ mixing was calculated using $(R_{xyA}(T)-R_{xyB}(T))/2$. Here $R_{xyA}$ and $R_{xyB}$ are the remanence with 1 and $-1$ T saturation, respectively.

**Data availability**. The data that support the findings of this study are available from the corresponding author upon request.

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

## Acknowledgements

This research was supported by the Australian Research Council Centre of Excellence in Future Low-Energy Electronics Technologies (Project no. CE170100039), the Institute for Information & Communications Technology Promotion (IITP) grant (Project no. B0117-16-1003, fundamental technologies of 2D materials and devices for the platform of new-functional smart devices), and the Basic Science Research Program (Project no. 2016R1A2B4012931), and the National Research Foundation (NRF) of Korea by a grant funded by the Korean Ministry of Science, ICT and Planning (Project no. 2012R1A3A2048816).

## Author contributions

C.L. and L.W. conceived and designed the research. J.L. synthesized the material, J.L., S.-G.J. and T.P. performed the material characterization. M.R.F. and D.G.M. performed the TEM scan for the cross-section of nanoflakes. C.T., J.P. and S.A. fabricated the Hall bar devices. C.T. and L.W. performed the electron transport measurements, data analysis and modeling. C.T., J.P., L.W., J.L. and C.L. wrote the paper with the help from all of the other co-authors.

## Additional information

**Competing interests:** The authors declare no competing interests.

