## [Peer Review File · Nature Communications]

Reviewers' comments:

Reviewer #1 (Remarks to the Author):

Referee's report on "Hard magnetic properties in nanoflake van der Waals Fe₃GeTe₂," by C. Tan, et al.

This manuscript reports thickness-dependent magnetic properties of Fe₃GeTe₂ inferred from anomalous Hall effect measurements. In contrast to the bulk, nanoflakes exhibit a hard magnetic phase with a square hysteresis loop and large coercivity.

The discovery of a metallic van der Waals magnet with a large magnetic remanence to saturated magnetization ratio is important for the development of functional van der Waals heterostructures and spintronic devices. For this reason, this paper is potentially important. However, the authors suggest that at thicknesses below 7 nm the properties of Fe₃GeTe₂ may change dramatically due to a 3D to 2D crossover.

The paper is well motivated and well referenced. It is clearly written for the most part, with the exception of the Stoner-Wohlfarth modeling section, which is a bit confusing and could be improved. For example, the model assumes a single domain, which flips at once, but then there is discussion about the first domain flipping. Is the first domain the whole sample? It is not clear. Also, the authors should make clear exactly what the Stoner-Wohlfarth model explains. Is it simply the angular dependence of the coercivity, or does it explain more?

Although I think the results in this paper are exciting and potentially important, I am not yet convinced that the authors have all experimental variables under control.

1) The most important issue that needs to be addressed is to confirm that the samples do not have an amorphous oxide layer growing on top. As the surface area to volume ratio grows with decreasing sample thickness, the increased coercivity could be explained by the increased influence of the surface layer on the thinner samples. I understand that the authors went to some lengths to minimize exposure of the samples to ambient conditions, but a significant oxide layer could still form quickly. I would like to see cross-sectional electron microscopy images. An example of what is needed can be found in Fig. 1 of <http://xxx.lanl.gov/abs/1505.03769>.

The other concerns I have with the manuscript are relatively minor.

- 1) Why is T_c lower than previously reported samples?
- 2) Is Fig. 1a really consistent with Fig. S1c?
- 3) The jump at 150 K in the magnetization needs to be understood better. Iron deficient samples show a transition near 150K. Could there be iron deficient regions in the sample, perhaps at the surface?
- 4) In Fig. S7a the resistivity rises as T is decreased--what is going on? Is the sample still a metal?
- 5) Demagnetization effects should be significant, but there was no discussion of correcting the applied field to account for these effects.
- 6) There is a statement in the paper: "When the temperature is higher than 150 K, the coercivity becomes zero because the thermal agitation energy is higher than the perpendicular anisotropic energy." However, in the Stoner Wohlfarth analysis section the anisotropy energy is estimated to be 25 kT. How can the thermal energy be larger? This is confusing.
- 7) Some information on the contact resistance should be provided. Also, why cannot resistivity be reported instead of just resistance?
- 8) It looks like some of the hysteresis loops are not symmetric and show what look like exchange bias effects. Can the authors comment?

Reviewer #2 (Remarks to the Author):

Comments:

With the addition of Fe_3GeTe_2 in the collection of van der Waals materials, the portfolio of possible two-dimensional magnets has further diversified to include now a ferromagnetic metal. As the authors state, a ferromagnetic metal will pave way for a completely new set of magnetic vdW heterostructures, potentially giving rise to new discoveries in not just two-dimensional materials, but in condensed matter physics as a whole. However, there are a long list of concerns, grammatical and content-related, that need to be addressed, before any further consideration for publication:

1. There is already a paper out on MBE-grown wafer-scale Fe_3GeTe_2 that demonstrated the magnetic properties in the ultrathin regime. This work (Liu et al. npj 2D materials, 2017)) needs to be referenced.

2. The lack of work on magnetic 2D materials isn't due to a scarcity in "desirable magnetic properties." There are a host of other issues such as stability of the material in ambient environments, proper experimental equipment, general interest, etc. The referee highly suggests that the authors remove this claim.
3. There is mention of scaling analysis, but this was never used in any form of analysis of the measurements. There are criticality fits that the authors performed, but those are not called "scaling analysis." Once again, the referee highly suggests that the authors re-phrase the abstract to avoid confusion about the actual analysis that was done to find the spin-spin correlation lengths.
4. The authors claim that thickness-dependent domain structures are what lead to the change in MR/MS and coercivity of a material, but they don't explain in detail how the domain structure changes in order to increase the two quantities. Please provide a citation or a detailed explanation for why thickness-dependent domain structures dramatically alter MR/MS and the coercivity and why this also occurs in ultrathin Fe₃GeTe₂.
5. What are the origins of this "soft" and "hard" magnetic phase that is seen in the 191 nm and the 329 nm flakes? What evidence do the authors have to show that these two phases are actually coexisting and not competing? For instance, other magnetic field transitions give rise to the same M vs. B dependence like spin-flop transitions in antiferromagnets?
6. In Huang et al. (Nature, 2017) and Zhong et al. (Science Advances, 2017), they clearly show the formation of domains in atomically-thin CrI₃. Surely, thin Fe₃GeTe₂ also exhibits multiple domains as well? Arguably, the 191 nm flake seems to have two domains demonstrated by a two-jump hysteresis loop rather than a one-jump loop. What evidence do the authors have to show that their ultrathin Fe₃GeTe₂ flakes are indeed single-domain?
7. It appears the coercivity increases as the thickness decreases from 82 nm down to 10.4 nm. Have the authors seen devices that break this trend, i.e. a 50 nm flake that has a smaller coercivity than the 82 nm flake? If not, what is causing this thickness-dependent coercivity to appear, especially since spin-spin correlation lengths decrease as thickness decreases?
8. The Curie temperature cannot be extracted using an unmodified Curie-Weiss law. This assumes that the critical exponent is 1 for the temperature-dependent susceptibility, which the authors show no evidence for. A simpler method to determine TC is to find the temperature at which the remanent R_{xy} signal goes to zero. In fact, it is this criterion that distinguishes a ferromagnet from a paramagnet. This is the approach that Gong et al. (Nature, 2017) took in searching for ferromagnetism in atomically-thin CrGeTe₃.
9. The authors claim 7.6 nm or 6.8 monolayers to be the critical thickness at which Fe₃GeTe₂ switches from 3D ferromagnetism to 2D ferromagnetism. Do the authors have atomic force microscopy data to support that monolayer is indeed ~1 nm? From TEM studies of Fe₃GeTe₂ done by Shan et al. (2017), they clearly show the single layer thickness to be about 0.82 nm. This will further lower the spin-spin correlation length to ~6 nm, so it isn't surprising that the authors did not see 2D magnetism in 7.6 nm-thick Fe₃GeTe₂.
10. The authors mention that neutron scattering measurements are needed in the future to resolve

the spike in remanence from 2 to 20 K. Unfortunately, reliable measurements are done on large, bulk crystals, so there may not be any scattering on a micron-scale 10 nm thick flake. How do the authors circumvent this problem?

11. There is use of mean-field theory as well as spin-wave theory to fit temperature-dependent R_{xy} data in fig. 3c, yet the authors do not detail how they implemented these fits. The referee highly suggests that the authors add a section in the supplementary detailing these calculations.

12. Although the authors had success in fitting their data with a modified Stoner-Wohlfarth model, this model would not explain the coexistence of multiple domains at a non-zero applied field, as seen on monolayer CrI₃ in Huang et al. (Nature, 2017). The referee highly suggests the authors to weaken their claim on this model being generalized to “other vdW ferromagnetic systems with strong perpendicular anisotropy.”

13. The authors state that $\theta = 85^\circ$ gave the best fit to the Stoner-Wohlfarth model. Why was this? How far off were the fits at smaller angles, like 20° ? It seems that even $\theta = 85^\circ$ has deviation at higher field. Why is that?

14. The explanation for why $a(T)$ decreases as a function of temperature in supplementary section 4.2 is counter-intuitive. The difference, $\theta - \phi$, denotes the canting of the magnetization vector away from the easy axis, and it's clear from experimental results that this canting is increasing to lower M_s as T increases. Why then do the author claim that $\theta - \phi$ lowers as temperature increases?

15. Looking over the reference list, numbers 1-10 are cited incorrectly. For instance, the isolation of graphene should of course, cite Novoselov et al. (2004) and Zhang et al. (2005), not five different papers in the 2010s on vdW heterostructures!

Until these problems have been addressed, the referee cannot recommend this article for publication.

Reviewer #3 (Remarks to the Author):

The authors find that Fe₃GeTe₂ material, which in bulk state is a ferromagnet with low coercivity and remnant magnetization, becomes a hard magnet with a very sharp magnetization switch, close to rectangular magnetization loop, and substantial coercivity when it is in a flake form, on the order of ten nm to a hundred(s) nm, by measuring Hall resistance R_{xy} . They use a Stoner-Wohlfarth (SW) model to fit the M_z , or alternatively R_{xy} . They show the fits to the R_{xy} data collected with magnetic field 85 degrees away from the direction perpendicular to the surface of the flake, to demonstrate that the sample acts as a single moment, i.e. it is a single domain FM, away from the transition region, and extract anisotropy parameter K_a . The authors then develop a modified SW model to treat the transition region around the coercivity field, and use it to extract such parameters as the volume of the first domain that flips its magnetization, and an interaction parameter between the moment of this domain and the rest of the sample's

moment pointing still in the opposite direction. This model describes the coercivity of the 85 deg data at different temperatures very well.

There are also some interesting unexpected features discovered, such as abrupt jump of remnant magnetization nearly to zero at 150 K, as well as additional appearance of extra remnant magnetization below 10 K. These results will be of value to the community dealing with magnetic phenomena.

On the general statement of FGT utility as a technological material, I disagree with the authors. Such utility is rather limited due to the low FM transition temperature T_c , well below room temperature. One has to go even further into the region II of the phase diagram, below 150 K, to make use of a high remnant field, sharp magnetization transition, and high coercivity of FGT. However, FGT provides as a good model system to study effects of shape anisotropy of thin samples and thickness dependence on enhancing hardness of magnetic properties of the FM materials.

Overall the data are interesting, and a model describing the transition region and coercivity would be an important advance. It will be of interest to the community of magnetism and magnetic phenomena. However, in my view the contribution does not rise up to the criteria for Nature Communications of broad enough generality of the phenomena discussed and its connection to other science areas. In addition there are some questions and interesting features in the data that I feel were not discussed or explained in sufficient depth warranted by the data. I therefore recommend against publication in Nature Communications.

There are several suggestions and questions that are offered below. Some of them relate to care taken in the manuscript preparation, some to some errors/omissions that might be caught during the editorial process. Some to consistency between the text, figures, and captions. Overall, the authors could have put more care in preparation of the manuscript.

What is characteristic ferromagnetic domain size of bulk material? Is it comparable to the thickness of the onset of the hard magnet behavior?

How well does the SW model describe the coercivity data obtained for other (than 85 deg) field directions? Perhaps calculations for few more directions can be shown in the Supplementary Materials.

p. 3 why “ M_z vs B loop when the field is perpendicular the sample surface”? Isn’t the same relation between M_z and R_{xy} used to produce fig. 4c?

p. 4. Hard magnetic phase vs soft magnetic phase – both coercivity fields are of the same order of magnitude, varying by roughly by a factor of two. So they are roughly of the same type – either both soft or both hard.

p. 8. State 1 is strictly a metastable state.

Fig. 1f. In the thinnest sample the magnetization close to the coercivity field (or transitions between the positive and negative Hall resistance) are clearly more rounded than in thicker samples. What can be causing that? Can this be reproduced by the modified SW model?

Fig. 3(c) Why reduced temperature scale is used, while the text does not mention the reduced

temperature values at all, and states the absolute temperature values, and it is left to a reader to translate between the two scales? why the dashed line drawn at $T/T_c = 0.1$, or $T = 20$ K? the kink in the M_T seems to occur close to 10 K. Also, the reduced M_T/M_0 is greater than 1 at $T=0$. So definition in the text: “ M_0 is the magnetic moment at 0 K” is strictly incorrect, or imprecise. M_0 is probably an extrapolation to $T=0$ from region II. If it is, this should be stated either in the text or in the caption. Also, the label on the vertical axis is magnetization, but the caption says “ R_{xy} ”. Please be consistent.

Fig. 4(c) caption: the deviation of the data from the fits to the SW model at high field are hard to attribute to the normal component, since the deviation is clearly non-linear, while the normal Hall resistance is linear in field (usually). Do the authors imply that the normal Hall resistance in FGT flakes is non-linear in field?

When coercive field increases with the field tipping angle off the perpendicular direction – the effect is not just the reduction of the perpendicular component of the magnetic field? What is the effect of the field component parallel to the plane of the flake, does it suppresses coercivity? How strong is this effect?

p. 5 Scaling behavior of FGT switching from 3d to 2d at 7.6nm? more than one point is needed to determine the scaling behavior for thickness below 7.6 nm. The statement “which is not shown in our figure” is ambiguous and therefore confusing. Is the meaning “the data do not show this behavior” or “we do not have the data in this regime”? or something else?

p. 6 What is the origin of the anisotropy in a flake of FGT? Crystalline or shape/geometric? What is the physical meaning of K_A in the experiment? does it matter? In either case I did not notice this discussion in the manuscript.

Do the values of the fitting parameters $a(T)$ and $V(T)$ depend on the choice of the coefficient of kT in the modified SW model? If they do, what is their utility? To show the trend with temperature and angle? Is it important to have a good handle on the size of the first flipping domain?

A question about the model, I hope the authors will clarify it for me. Equation (5) and similar equations in the Supplementary Materials: eq (5) describes the energy difference between states 1 and 2 for a first flipping domain. This domain, when in the state 1, is just a part of the old single domain state. Why its energy should not be just a scaled energy of the single domain state, i.e. $E(\text{domain in state 1}) = KaV(T)\sin^2(\varphi_2 - \theta) - V(T)M_s B \cos(\varphi_2)$? Or is it necessary in this case to include the energy of the ferromagnetically ordered sample, as it changes in state 2? Is interaction parameter needed to describe the energy of the first domain in the old single domain state? For the first domain that flips its energy in the excited (barrier) state 2, shouldn't it be $E(2) = KV(T)\sin^2(\varphi_2 - \theta) - V(T)MB \cos(\varphi_2) + a(T)V(T)M_s * M_s * \cos(\varphi_1 - \varphi_2)$ or something like that? Should there be another similar term in the expression for the small domain in state 1, such as $a(T)V(T)M_s * M_s$?. Why interaction energy between the first flipping domain and the rest of the sample with old magnetization depends on the applied magnetic field? Also, the angle φ_2 is calculated for the single domain case, when magnetization of a whole sample is φ_2 away from the field direction. How can we be sure that when a single domain flips, its barrier (highest

energy) state will have the same angle as ϕ_2 , for when the whole sample's magnetization rotates? Should the energy be $E(3) = KV(T)\sin^2(\phi_3 - \theta) - V(T)MB\cos(\phi_3) + a(T)V(T)M_s M_s \cos(\phi_1 - \phi_3)$, where ϕ_3 may not be equal to ϕ_2 ?

Supplementary Material:

fig S5(f) should the symbols be reversed?

Fig. S5(a) are the 120K shown? It looks like the 150 K data's "hard phase" has the same coercivity as 100 K data.

Fig. S7(b). the figure caption is confusing: (b) Temperature dependence of remanence and R_{xy} curve with 1 T applied magnetic field. Is R_{xy} at 1T labeled as saturated value in the figure? A bit more clarity and consistency between figure and a caption would of value.

The discussion below (at least a condensed and/or selective portion of it, see italicized part) would be commonly placed in Methods and Materials section of the manuscript, instead of supplementary materials:

"Because of the non-symmetry in our nanoflake devices, the measured Hall resistance was mixed with the longitudinal magnetoresistance. We processed the data by using $(R_{xyA}(+B) - R_{xyB}(-B)) / 2$ to eliminate the contribution from the longitudinal magnetoresistance, where R_{xyA} is the half loop sweeping from the positive field to the negative field, R_{xyB} is the half loop sweeping from the negative field to the positive field, and B is the applied magnetic field. We also measured the $R_{xy}(T)$ at the remanence point for all the samples. To measure the $R_{xy}(T)$ at the remanence point, the magnetic moment of samples was first saturated by a 1 T magnetic field and then the magnetic field was decreased to zero (the remanence point). Finally, the temperature dependence of the R_{xy} at remanence was measured when the temperature was increased from 2 K to 300 K. In order to eliminate the non-symmetry effect of the device, we measured R_{xy} (remanence) with both 1 T and -1 T saturation. The real $R_{xy}(T)$ at remanence without R_{xx} mixing was calculated using $(R_{xyA}(T) - R_{xyB}(T)) / 2$. Here R_{xyA} and R_{xyB} are the remanence with 1 T and -1 T saturation, respectively."

How do we know that the procedure described above for measuring remanence as a function of temperature is not effected by hysteretic effects between field and temperature sweeps? Were at least a couple of points checked and consistency established? If so, it would be helpful to put some of the field-sweep remanence points on the "temperature dependent remanence" figures.

Reviewer #1 (Remarks to the Author):

Comments:

Referee's report on "Hard magnetic properties in nanoflake van der Waals Fe_3GeTe_2 ," by C. Tan, et al.

This manuscript reports thickness-dependent magnetic properties of Fe_3GeTe_2 inferred from anomalous Hall effect measurements. In contrast to the bulk, nanoflakes exhibit a hard magnetic phase with a square hysteresis loop and large coercivity.

The discovery of a metallic van der Waals magnet with a large magnetic remanence to saturated magnetization ratio is important for the development of functional van der Waals heterostructures and spintronic devices. For this reason, this paper is potentially important. However, the authors suggest that at thicknesses below 7 nm the properties of Fe_3GeTe_2 may change dramatically due to a 3D to 2D crossover.

The paper is well motivated and well referenced. It is clearly written for the most part, with the exception of the Stoner-Wohlfarth modeling section, which is a bit confusing and could be improved. For example, the model assumes a single domain, which flips at once, but then there is discussion about the first domain flipping. Is the first domain the whole sample? It is not clear. Also, the authors should make clear exactly what the Stoner-Wohlfarth model explains. Is it simply the angular dependence of the coercivity, or does it explain more?

Reply: We thank the reviewer for the valuable comments. Our model is a **modified Stoner-Wohlfarth model**. The traditional Stoner-Wohlfarth model describes a single domain system in the whole range of magnetic field, while Fe_3GeTe_2 behaves as a single domain system **in the magnetic field regime away from the coercivity**. Therefore, model in this

paper still needs to consider the first domain flipping happens around the coercivity. In a narrow regime of magnetic field near the coercivity, domains appear and induce the flip of magnetic moments. Figure R1 shows the single domain regime (the blue curve) and the multi-domain regime (the red curve). The blue curve can be fitted well by the model of single domain, from which the very important **perpendicular anisotropic energy (K_A)** can be obtained.

Using the modified Stoner-Wohlfarth model, the experimental angular dependence of coercivity can be well fitted. From the fitting, the size of the first flipped domain and the dragging effect from unflipped domains can be estimated. These two parameters are useful for understanding the dynamics of the system and designing spin orbital torque devices based on Fe_3GeTe_2 .

Fig. R1 Schematic of the single- and multi- domain regimes in the anomalous Hall effect curves measured at different tilt angles.

Revision:

We revised the description of the model to make it clearer and easier for understanding.

Comments:

Although I think the results in this paper are exciting and potentially important, I am not yet convinced that the authors have all experimental variables under control.

1) The most important issue that needs to be addressed is to confirm that the samples do not have an amorphous oxide layer growing on top. As the surface area to volume ratio grows with decreasing sample thickness, the increased coercivity could be explained by the increased influence of the surface layer on the thinner samples. I understand that the authors went to some lengths to minimize exposure of the samples to ambient conditions, but a significant oxide layer could still form quickly. I would like to see cross-sectional electron microscopy images. An example of what is needed can be found in Fig. 1 of <http://xxx.lanl.gov/abs/1505.03769>.

Reply: The cross-sectional TEM image is shown in Figure R2a. We choose the same time of ambient exposure as that in our device fabrication and transport measurements (7 mins). The image shows that there is indeed an amorphous oxide layer of ~1.2 nm thick on the sample.

To check whether FGT flakes still show hard magnetic phase with a near square-shaped loop without an oxide layer, we fabricated ultra-clean devices using our new vdW fabrication system in a glove box with both O_2 and $H_2O < 0.1$ ppm. The image of one of the devices is shown in Fig R2c. To fabricate this device, we utilized the method in [¹*Nature Physics* **13**, 677–682 (2017)]. Firstly, 5 nm thick Pt contacts were fabricated on Si/SiO_x substrate in ambient condition. In our glove box, an exfoliated FGT flake was then dry transferred onto the contacts to form very good ohmic contacts. Thereafter, a large hBN flake was dry transferred to cover the FGT flake to prevent oxidization during measurements. Finally, the sample was covered by PMMA to prevent any possible oxidization. The R_{xy} vs H loop is shown in Fig R2d. It is very clear that ultra clean FGT flakes still show hard magnetic property with square shaped loop. The $R_{xy}(T)$ curve of the device also shows the same characteristics as that of the devices in main text. Thus, the thin

Fig R2 (a) Cross-sectional TEM image of an FGT nanoflake on substrate. The top layer is 5 nm Pt layer. The oxide layer is about 1.2 nm. The thickness of monolayer is about 0.8 nm. The scale bar is 10 nm. (b) Diffraction pattern of the FGT nanoflake. (c) A 5.6 nm FGT device covered by hBN, the bottom contact is 5 nm Pt. The red dashed line is the FGT region, yellow dashed line is the h-BN region. The scale bar is 10 μ m. (d) Anomalous Hall effect at 2 K. Magnetic field is perpendicular to the sample surface.

oxide layer on the sample surface does not affect the main conclusions (hard magnetic properties with a near square-shaped loop) of this paper. We can also see that FGT is a promising material whose magnetism can survive in ambient environment for a certain time. From our experiments, we conclude that the effect of oxide layer includes:

1. The switch of magnetic moment in the square shaped loop of FGT with oxide layer is not as sharp as that in ultra-clean FGT flakes, which is due to the pinning effect of the oxide layer.
2. The coercivity of FGT slightly increases after the oxidization, which is also due to the domain wall pinning effect.

Revision: A new section discussing the effect of oxidization has been added in supplementary information. The T_c s of two ultra-clean devices has been added in Fig. 2.

Comments:

The other concerns I have with the manuscript are relatively minor.

Comments: 1) Why is T_c lower than previously reported samples?

Reply: Previously, T_c of Fe_3GeTe_2 crystal was mostly reported to be around 220 K. The T_c of our bulk crystal is around 205 K, which is a bit lower than the previous cases as pointed out by the reviewer. A recent report² on Fe_3GeTe_2 crystals studying the iron composition effect on T_c showed that deficiency of iron atoms in the crystal lowers the transition temperature. We performed EDS analysis to find the composition ratio between the elements, and the elemental ratio was Fe:Ge:Te=2.88:1:2.05, which reveals the iron deficiency in the crystal. Hence, we consider that the iron deficiency is the most probable cause of the lower T_c in this crystal.

Fig. R3 EDS analysis of a bulk Fe_3GeTe_2 single crystal.

Comments: 2) Is Fig. 1a really consistent with Fig. S1c?

Reply: We replot the Fig. 1a and Fig. S1c to Fig. R2. It can be seen the main features of of Fig. 1a and Fig. S1c are consistent. These include, very small coercivity (100 Oe~ 200 Oe), very small remanence, and the shape of the magnetic loop. As we used different crystals

Fig. R4 (a) Magnetic measurement on bulk FGT single crystal. (b) Electric measurement on bulk FGT single crystal.

for transport measurements (Fig. 1a) and magnetic measurements (Fig. S1c), there should be some differences in coercivity and saturation in two measurements. The relative orientation between the magnetic field and crystalline direction may also have some differences in the two experiments.

Comments: 3) The jump at 150 K in the magnetization needs to be understood better. Iron deficient samples show a transition near 150K. Could there be iron deficient regions in the sample, perhaps at the surface?

Reply: Previous report on bulk FGT crystals³ indicates a phase transition at 150 K for iron stoichiometric crystals. We speculate that the sharp transition at 150 K may not be a special phenomenon only for iron deficient samples. The sharp transition may be related to decreased thickness of nanoflakes, which induces single domain behaviour at the remanence.

Revision: We add a discussion of the sharp transition at 150 K in the revised manuscript. "As bulk FGT single crystal also shows phase transition with gradually changed magnetic moment around 155 K, we speculate that the sharp transition in FGT nanoflake is due to the decreased thickness which induces single domain behaviour at the remanence."

Comments: 4) In Fig. S7a the resistivity rises as T is decreased--what is going on? Is the sample still a metal?

Reply: The accurate definition of metal is "materials show finite resistance at zero temperature ". Based on this definition, the FGT nanoflake is a metal. In many situations, a metal can show an enhanced resistivity with decreasing temperature. Disorders in materials can induce Anderson localization. If the Fermi level is in the localized state, the material is still a metal but it will show enhanced resistance with decreasing temperature.

The resistivity of 10.4 nm nanoflakes as shown in Fig S7 is in the order of $10^{-4} \Omega \cdot \text{cm}$, which is definitely a metal. The enhanced resistance with decreasing temperature may originate from the disorder at the sample surface induced by exfoliation or oxidation.

Revision: We add explanation in the caption of Fig S7. "The enhanced resistance with decreasing temperature may originate from the disorder at the sample surface induced by exfoliation or oxidation."

Comments: 5) Demagnetization effects should be significant, but there was no discussion of correcting the applied field to account for these effects.

Reply: We thank the reviewer#1 for pointing out this important issue. Based on this comment, we considered the demagnetization effects in FGT system. Based on previous research⁴, for a thin film magnetic system with strong perpendicular anisotropy, the demagnetization parameter $D \cong 1$ and $D \cong 0$ along in-plane direction and out of plane

Fig. R5 (a) Fittings of data with and without considering demagnetization. (b) Angle dependence of magnitude of the applied magnetic field and effective magnetic field at 2 K.

direction, respectively. With the modification of demagnetization effect, we obtained better fittings of magnetic loop as shown in Fig. R5a. The values of the angle and magnitude of the applied magnetic field and effective magnetic field at 2 K are shown in Fig. R5b.

Revision: All fittings in this paper have been done again. A detailed description of the demagnetization effect is added in the supplementary materials.

Comments: 6) There is a statement in the paper: "When the temperature is higher than 150 K, the coercivity becomes zero because the thermal agitation energy is higher than the perpendicular anisotropic energy." However, in the Stoner Wohlfarth analysis section the anisotropy energy is estimated to be $25 k_B T$. How can the thermal energy be larger? This is confusing.

Reply: The description here is a very rough explanation. Based on the modified Stoner-Wohlfarth model, the magnetization starts to flip when the $K_A \sim 25 k_B T$.

Revision: We delete the parts of the sentence in the revised manuscript. "When the temperature is higher than 150 K, the coercivity becomes zero."

Comments: 7) Some information on the contact resistance should be provided. Also, why cannot resistivity be reported instead of just resistance?

Reply: For the R_{xx} measurement in the supplementary information, the resistivity is very important as pointed by reviewer #1.

For the R_{xy} data, just like MOKE measurements, we cannot get the value of magnetization from the anomalous Hall measurements. The value R_{xy} usually can be written as "arbitrary units". Hence, for the R_{xy} in this paper, we think it does not matter whether to express it as resistance or resistivity.

Fig. R6 (a) R_{xx} vs Current curve at 2 K for an FGT sample. (b) Corresponding I-V curve derived from (a).

Revision: We replotted the figures of R_{xx} using resistivity as unit in supplementary information. An I-V curve of an FGT device is shown in Fig R6, which indicates ohmic contact. This figure is also plotted as Fig S11 in the supplementary materials.

Comments: 8) It looks like some of the hysteresis loops are not symmetric and show what look like exchange bias effects. Can the authors comment?

Reply: We did observe non-symmetric magnetic loops for few times when the angle $\theta > 70^\circ$ at $T < 10$ K, as shown in Fig. S4(c). This phenomenon looks like exchange bias. However, it is not easily repeatable like a normal exchange bias effect. Currently, we are trying to further confirm this interesting phenomenon. Hence, we only present these results in supplementary information without further discussion. One thing for sure is that the effect has no relationship with the oxide layer on the surfaces of FGT flakes.

Reviewer #2 (Remarks to the Author):

Comments:

With the addition of Fe_3GeTe_2 in the collection of van der Waals materials, the portfolio of possible two-dimensional magnets has further diversified to include now a ferromagnetic metal. As the authors state, a ferromagnetic metal will pave way for a completely new set of magnetic vdW heterostructures, potentially giving rise to new discoveries in not just two-dimensional materials, but in condensed matter physics as a whole.

Reply: We thank reviewer #2 for pointing out the importance of this work.

Comments: However, there are a long list of concerns, grammatical and content-related, that need to be addressed, before any further consideration for publication:

1. There is already a paper out on MBE-grown wafer-scale Fe_3GeTe_2 that demonstrated the magnetic properties in the ultrathin regime. This work (Liu et al. npj 2D materials, 2017)) needs to be referenced.

Revision: The paper of Liu et al. has been referenced in the new version of this paper.

2. The lack of work on magnetic 2D materials isn't due to a scarcity in "desirable magnetic properties." There are a host of other issues such as stability of the material in ambient environments, proper experimental equipment, general interest, etc. The referee highly suggests that the authors remove this claim.

Revision: Based on reviewer #2's suggestion, we have removed this claim in the new version of the paper.

3. There is mention of scaling analysis, but this was never used in any form of analysis of the measurements. There are criticality fits that the authors performed, but those are not called "scaling analysis." Once again, the referee highly suggests that the authors re-phrase the abstract to avoid confusion about the actual analysis that was done to find the spin-spin correlation lengths.

Revision: Based on the reviewer #2's suggestion, we have rephrased the abstract.

4. The authors claim that thickness-dependent domain structures are what lead to the change in M_R/M_S and coercivity of a material, but they don't explain in detail how the domain structure changes in order to increase the two quantities. Please provide a citation or a detailed explanation for why thickness-dependent domain structures dramatically alter M_R/M_S and the coercivity and why this also occurs in ultrathin Fe_3GeTe_2 .

Reply: Chapter 16 in "Modern Magnetic Materials: Principles and Applications"⁵ by O'Handley provides very detailed description on the differences of domain structures in bulk and thin film. Fundamental magnetic properties depend on the local environment: the symmetry, number, type and distance of atom's neighbours. The symmetry at the surface is radically altered relative to the bulk. Also, the number of nearest-neighbour atoms changes relative to the bulk. More

simply, when the thickness of a nanoflake becomes smaller than the dimension of the magnetic domain in bulk state, the domain structure will definitely change. The shapes of magnetic loops are determined by domain structure and the dynamics of domain structure. “*Hysteresis in magnetism: for physicists, materials scientists, and engineers.*”⁶ by Giorgio provides very detailed derivation and description. The paper in [Physical Review B vol 58, 3223] also provides a detailed analysis on a ferromagnetic thin film with perpendicular anisotropy. Fig. 3 in this paper shows a magnetic phase diagram of a ferromagnetic thin film with perpendicular anisotropy.

Revision: Based on the suggestion of reviewer #2, “Modern Magnetic materials: Principles and Applications” by O’Handley, “Magnetic Hysteresis” by Giorgio and Physical Review B vol 58, 3223 are added in the references.

5. What are the origins of this “soft” and “hard” magnetic phase that is seen in the 191 nm and the 329 nm flakes? What evidence do the authors have to show that these two phases are actually coexisting and not competing? For instance, other magnetic field transitions give rise to the same M vs. B dependence like spin-flop transitions in antiferromagnets?

Reply: “Two phase” behaviours of magnetic loops with different origins are very common in various magnetic systems. To fully understand the “two phase” behaviour in FGT system, more experiments, such as magnetic force microscopy (MFM) under applied magnetic field at low temperature, are required, which is beyond the scope of this paper.

However, with decreasing thickness from bulk to several nanometers, FGT gradually evolves from a soft phase (bulk) to a single hard phase. Based on this phenomenon, we speculate that the “two phase” behaviour is due to the gradual evolution of the domain structure.

Revision: We have revised the manuscript in page 4. “As FGT gradually evolves from a soft phase (bulk) to a single hard phase (82 nm, 49 nm and 10.4 nm), we speculate that the “two phase” behaviour in the nanoflakes with thicknesses of 329 nm and 191 nm is due to the gradual evolution of the domain structure.”

6. In Huang et al. (Nature, 2017) and Zhong et al. (Science Advances, 2017), they clearly show the formation of domains in atomically-thin CrI_3 . Surely, thin Fe_3GeTe_2 also exhibits multiple domains as well? Arguably, the 191 nm flake seems to have two domains demonstrated by a two-jump hysteresis loop rather than a one-jump loop. What evidence do the authors have to show that their ultrathin Fe_3GeTe_2 flakes are indeed single-domain?

Reply: Our sample is not a purely single domain system. As is clarified in our paper, FGT shows a single domain behaviour in the magnetic field regime **away from the coercivity**. Magnetic domain appears in a narrow regime near the coercivity. This behaviour is the same as that of atomically-thin CrI_3 .

In the regime away from the coercivity, we can use the standard Stoner-Wohlfarth model to fit the hysteresis loop to obtain perpendicular magnetic anisotropy, which is shown in Fig 4c and Fig. S7(d-h). To describe the angular dependence of coercivity, we proposed a simple modified Stoner-Wohlfarth model (Eq. 5). The V in the modified Stoner-Wohlfarth model is the volume of the first flipping domain. As the magnetic transition is pretty sharp for FGT, all the approximations in this

simple modified Stoner-Wohlfarth model are reasonable and the model can fit the angular dependence of coercivity very well.

As ultrathin CrI₃ has a similar square-shaped magnetic loop and perpendicular magnetic anisotropy, our simple model should also work well for CrI₃.

Revision: We revised the manuscript to make the description of our model clearer and easier to understand.

7. It appears the coercivity increases as the thickness decreases from 82 nm down to 10.4 nm. Have the authors seen devices that break this trend, i.e. a 50 nm flake that has a smaller coercivity than the 82 nm flake? If not, what is causing this thickness-dependent coercivity to appear, especially since spin-spin correlation lengths decrease as thickness decreases?

Reply: Up to now, more than 100 various spintronic devices based on FGT nano-flakes have been fabricated in Lan Wang's group. Based on all the measurement results, we can conclude that there is no clear relationship between the coercivity and the thicknesses in the range from 100 nm to 10 nm. We can observe smaller coercivity in relatively thinner flakes. The device fabrication process can also affect the coercivity of an FGT flake. For example, a short scan of FIB (Ga source, 30 kV, 1.5pA) can decrease the coercivity of an FGT flake. Hence we believe that defects, stoichiometry and the thin oxide layer on FGT surface are more important factors comparing with thickness in the range of 100 nm to 10 nm for the control of coercivity.

8. The Curie temperature cannot be extracted using an unmodified Curie-Weiss law. This assumes that the critical exponent is 1 for the temperature-dependent susceptibility, which the authors show no evidence for. A simpler method to determine T_C is to find the temperature at which the remanent R_{xy} signal goes to zero. In fact, it is this criterion that distinguishes a ferromagnet from a paramagnet. This is the approach that Gong et al. (Nature, 2017) took in searching for ferromagnetism in atomically-thin CrGeTe₃.

Revision: Based on this comment, we determined the T_C using the suggested method and refitted the thickness dependence of T_C . Related parts in supplementary information and Fig. 2 have been revised accordingly.

9. The authors claim 7.6 nm or 6.8 monolayers to be the critical thickness at which Fe₃GeTe₂ switches from 3D ferromagnetism to 2D ferromagnetism. Do the authors have atomic force microscopy data to support that monolayer is indeed ~1 nm? From TEM studies of Fe₃GeTe₂ done by Shan et al. (2017), they clearly show the single layer thickness to be about 0.82 nm. This will further lower the spin-spin correlation length to ~6 nm, so it isn't surprising that the authors did not see 2D magnetism in 7.6 nm-thick Fe₃GeTe₂.

Reply: We thank reviewer #2 for pointing this out. We do know that a single layer of FGT is about 0.82 nm. However, when we wrote the paper, we mixed the number of layers and the thickness value and made the mistake.

Revision: We revised the relative description in the abstract and main text.

10. The authors mention that neutron scattering measurements are needed in the future to resolve the spike in remanence from 2 to 20 K. Unfortunately, reliable measurements are done on large, bulk crystals, so there may not be any scattering on a micron-scale 10 nm thick flake. How do the authors circumvent this problem?

Reply: For neutron scattering measurements, we do need large samples. FGT thin films grown in a MBE system are suitable for this experiment. Another possible method is to grow nanoflakes or thin films using a vapour transport method (a tube furnace with Ar/H₂ gas flow). Using this method, a large area of a substrate (>50%) can be covered by FGT flakes, which is suitable for neutron scattering experiments.

11. There is use of mean-field theory as well as spin-wave theory to fit temperature-dependent R^{xy} data in fig. 3c, yet the authors do not detail how they implemented these fits. The referee highly suggests that the authors add a section in the supplementary detailing these calculations.

Revision: A new section about mean field fitting and spin wave fitting has been added in the supplementary information.

12. Although the authors had success in fitting their data with a modified Stoner-Wohlfarth model, this model would not explain the coexistence of multiple domains at a non-zero applied field, as seen on monolayer CrI₃ in Huang et al. (Nature, 2017). The referee highly suggests the authors to weaken their claim on this model being generalized to “other vdW ferromagnetic systems with strong perpendicular anisotropy.”

Reply: We partially agree with reviewer#2 on this point, the modified Stoner-Wohlfarth model can be used for vdW ferromagnetic materials with strong perpendicular anisotropy and showing square-shaped loop. CrI₃ is actually such a system. Traditional Stoner-Wohlfarth model describes a single domain system in the whole range of magnetic field, while our modified Stoner-Wohlfarth model describes a system behaving as a single domain system in the magnetic field regime away from the coercivity. In a narrow regime of magnetic field near the coercivity, domains appear and induce the flip of magnetic moments. Figure R7 shows the single domain regime (the blue curve) and the regime of multi-domain regime (the red curve). The blue curve can be fitted well by the model of single domain, from which the very important **perpendicular anisotropic energy** can be obtained.

Fig. R7 Schematic of the single- and multi-domain regimes in the anomalous Hall effect curves measured at different tilt angles.

Using the modified Stoner-Wohlfarth model, the experimental angular dependence of coercivity can be well fitted. From the fitting, the size of the first flipped domain and the dragging effect from unflipped domains can also be estimated. These two parameters are useful for understanding the dynamics of the system and designing spin orbital torque devices based on FGT or other vdW ferromagnets with strong perpendicular anisotropy and square-shaped loop.

13. The authors state that $\theta = 85^\circ$ gave the best fit to the Stoner-Wohlfarth model. Why was this? How far off were the fits at smaller angles, like 20° ? It seems that even $\theta = 85^\circ$ has deviation at higher field. Why is that?

Reply: Actually, the magnetic loops at all angles can be well fitted by Stoner-Wohlfarth model. We focus on the fitting of magnetic loop at $\theta = 85^\circ$ because the magnetic loop at $\theta = 85^\circ$ has more varying curvatures to obtain a more accurate fitting value of magnetic anisotropy energy (K_A). It is easier to get good fittings at lower angles, at which the curve is pretty much a horizontal line. However, there is a wide range of K_A values that can provide good fittings under this condition. Hence, we cannot get a reliable fit of K_A at small θ . For $\theta = 85^\circ$ curve, the best fitting can provide a reliable K_A . Moreover, based on reviewer #1's suggestion on demagnetization effect, we refitted all the curves, the fittings are actually much better according to the revised picture in Fig.S4c and Fig.S7(d-h).

14. The explanation for why $a(T)$ decreases as a function of temperature in supplementary section 4.2 is counter-intuitive. The difference, $\phi - \theta$, denotes the canting of the magnetization vector away from the easy axis, and it's clear from experimental results that this canting is increasing to lower M_s as T increases. Why then do the author claim that $\phi - \theta$ lowers as temperature increases?

Reply: The value of $\phi - \theta$ is determined by Eq. (4) (the Stoner-Wohlfarth model). The smaller the magnetic field is, the smaller the $\phi - \theta$. When the magnetic field is zero, $\phi - \theta$ is equal to zero and the magnetic moments align with the easy axis. At higher temperatures, the coercivity is smaller. Therefore, the $\phi - \theta$ is actually smaller due to the smaller coercive field at higher temperatures.

15. Looking over the reference list, numbers 1-10 are cited incorrectly. For instance, the isolation of graphene should of course, cite Novoselov et al. (2004) and Zhang et al. (2005), not five different papers in the 2010s on vdW heterostructures!

Revision: We have revised the relevant citations based on this comment.

Reviewer #3 (Remarks to the Author):

Comments: The authors find that Fe_3GeTe_2 material, which in bulk state is a ferromagnet with low coercivity and remnant magnetization, becomes a hard magnet with a very sharp magnetization switch, close to rectangular magnetization loop, and substantial coercivity when it is in a flake form, on the order of ten nm to a hundred(s) nm, by measuring Hall resistance R_{xy} . They use a Stoner-Wohlfarth (SW) model to fit the M_z , or alternatively R_{xy} . They show the fits to the R_{xy} data collected with magnetic field 85 degrees away from the direction perpendicular to the surface of the flake, to demonstrate that the sample acts as a single moment, i.e. it is a single domain FM, away from the transition region, and extract anisotropy parameter K_A . The authors then develop a modified SW model to treat the transition region around the coercivity field, and use it to extract such parameters as the volume of the first domain that flips its magnetization, and an interaction parameter between the moment of this domain and the rest of the sample's moment pointing still in the opposite

direction. This model describes the coercivity of the 85 degree data at different temperatures very well.

Reply: Reviewer #3 probably mis-understands our results as shown in the last sentence (the red part). Fig. 4d shows clearly that the modified SW model well describes the coercivity of FGT nanoflakes from 0° to 85° in the temperature regime from 2 K to 120 K.

There are also some interesting unexpected features discovered, such as abrupt jump of remnant magnetization nearly to zero at 150 K, as well as additional appearance of extra remnant magnetization below 10 K. These results will be of value to the community dealing with magnetic phenomena.

On the general statement of FGT utility as a technological material, I disagree with the authors. Such utility is rather limited due to the low FM transition temperature T_c , well below room temperature. One has to go even further into the region II of the phase diagram, below 150 K, to make use of a high remnant field, sharp magnetization transition, and high coercivity of FGT. However, FGT provides as a good model system to study effects of shape anisotropy of thin samples and thickness dependence on enhancing hardness of magnetic properties of the FM materials.

Reply: We disagree with reviewer #3 regarding the importance of this material. FGT nanoflake is actually the **FIRST vdW ferromagnetic metal** showing hard magnetic property with $M_R/M_S \sim 1$. The material can be easily exfoliated to nanoflakes with a thickness of several nanometers. Using a standard stacking method of vdW materials, various spintronic devices based on vdW heterostructures can be designed and fabricated and therefore new physics can be explored. For example, novel spin orbit torque devices can be fabricated using FGT based vdW heterostructures, which is very different from conventional spin orbit torque devices based on thin film growth technique. In this case, many experiments about spin orbit torque can be realized based on vdW heterostructure, which cannot be done in the normal thin film based spin orbit torque devices. Here, we just give two examples. 1. Using vdW heterostructures, we can easily control the relative crystalline orientation between the ferromagnetic layer and the paramagnetic spin orbit material. 2. We can insert a monolayer material with special properties between the ferromagnetic layer and the paramagnetic spin orbit material, which may induce spin orbit devices with ultra-low threshold current. This is extremely important for technological application. Many more other novel experiments can be designed based on the new vdW ferromagnetic material, FGT nanoflakes.

Overall the data are interesting, and a model describing the transition region and coercivity would be an important advance. It will be of interest to the community of magnetism and magnetic phenomena. However, in my view the contribution does not rise up to the criteria for Nature Communications of broad enough generality of the phenomena discussed and its connection to other science areas. In addition there are some questions and interesting features in the data that I feel were not discussed or explained in sufficient depth warranted by the data. I therefore recommend against publication in Nature Communications.

Reply: We thank the suggestions from the reviewer #3 to improve the paper, but disagree with reviewer #3 regarding the impact of the work. As aforementioned, because FGT is the first vdW ferromagnetic metal with hard magnetism and $M_R/M_S \sim 1$, this pioneering work makes the fabrication of various novel spintronic magneto-optic devices based on ferromagnetic vdW

heterostructures possible. The communities of magnetism and magnetic phenomena, 2D materials, spintronics will all be interested in this work.

There are several suggestions and questions that are offered below. Some of them relate to care taken in the manuscript preparation, some to some errors/omissions that might be caught during the editorial process. Some to consistency between the text, figures, and captions. Overall, the authors could have put more care in preparation of the manuscript.

Comments 1: What is characteristic ferromagnetic domain size of bulk material? Is it comparable to the thickness of the onset of the hard magnet behavior?

Reply: The characteristic ferromagnetic domain has been investigated in previous paper⁷ [J. Appl. Phys. 120, 083903 (2016) supplementary information]. The standard size is about $\sim 1 \mu\text{m} \times 1 \mu\text{m}$.

Comments 2: How well does the SW model describe the coercivity data obtained for other (than 85 deg) field directions? Perhaps calculations for few more directions can be shown in the Supplementary Materials.

Reply: As mentioned before, reviewer #3 misunderstands our model. Our model (the modified SW model) can well describe the angular dependence of coercivity from 0° to 85° . Using the SW model, we can well fit all the magnetic loops from (0° to 85°) in the magnetic field regime of single domain. We focus on the magnetic loops fitting at 85° , because we can obtain more accurate K_A (magnetic anisotropy) at this angle. If the angle is small, the loop will be pretty much a straight line. Hence, a wide range of K_A can provide good fitting when the angle is small, then the K_A value cannot be accurately fitted.

Comments 3: Why “ M_z vs B loop when the field is perpendicular the sample surface”? Isn’t the same relation between M_z and R_{xy} used to produce fig. 4c?

Reply: We thank reviewer #3 for pointing out this error. “the field is perpendicular to the sample surface” is not required, which shows in the whole content of this paper.

Revision: We have deleted “when the field is perpendicular to the sample surface.”

Comments 4: p. 4. Hard magnetic phase vs soft magnetic phase – both coercivity fields are of the same order of magnitude, varying by roughly by a factor of two. So they are roughly of the same type – either both soft or both hard.

Reply: We thank the reviewer#3 for pointing out this issue, the “soft” and “hard” magnetic phases are not accurate in the paper.

Revision: We revised the manuscript based on reviewer #3’s suggestion. We replace “soft” and “hard” with “two phases with different coercivity.”

Comments 5: p. 8. State 1 is strictly a metastable state.

Revision: We replace “stable state” with “metastable state”.

Comments 6: Fig. 1f. In the thinnest sample the magnetization close to the coercivity field (or transitions between the positive and negative Hall resistance) are clearly more rounded than in thicker samples. What can be causing that? Can this be reproduced by the modified SW model?

Reply: The rounded hysteresis loop near coercivity is due to the pinning of the domain wall expansion, which means that the sample has more defects. To reproduce this feature of a magnetic loop, micro-magnetic simulation is required. The modified SW model is a simple phenomenological model. It cannot reproduce this kind of phenomenon.

Comments 7: Fig. 3(c) Why reduced temperature scale is used, while the text does not mention the reduced temperature values at all, and states the absolute temperature values, and it is left to a reader to translate between the two scales? why the dashed line drawn at $T/T_c = 0.1$, or $T = 20$ K? the kink in the M_T seems to occur close to 10 K. Also, the reduced M_T/M_0 is greater than 1 at $T=0$. So definition in the text: “ M_0 is the magnetic moment at 0 K” is strictly incorrect, or imprecise. M_0 is probably an extrapolation to $T=0$ from region II. If it is, this should be stated either in the text or in the caption. Also, the label on the vertical axis is magnetization, but the caption says “ R_{xy} ”. Please be consistent.

Revision: We revised carefully based on Reviewer #3’s comments. Fig. 2c has been revised.

Comments 8: Fig. 4(c) caption: the deviation of the data from the fits to the SW model at high field are hard to attribute to the normal component, since the deviation is clearly non-linear, while the normal Hall resistance is linear in field (usually). Do the authors imply that the normal Hall resistance in FGT flakes is non-linear in field?

Reply: Review #1 pointed out that demagnetization effect should also be considered. Based on his comment, we considered the demagnetization effects in FGT system. Based on previous research⁴, for a thin film magnetic system with strong

perpendicular anisotropy, the demagnetization parameter $D \cong 1$ and $D \cong 0$ along in-plane direction and out of plane direction, respectively. With the modification of demagnetization effect, we obtained better fittings of magnetic loop as shown in Fig. R8a. The values of the angle and magnitude of the applied magnetic field and effective magnetic field at 2 K are shown in Fig. R8b. Based on the new fittings, we believe the effect of normal Hall effect is actually not as large as we thought.

Revision: All fittings in this paper have been done again. A detailed description of the demagnetization effect is added in the supplementary materials. The discussion about the contribution of the normal Hall effect component has also been deleted.

Fig. R8 (a) Fittings of magnetic loop with and without considering demagnetization. (b) Angle dependence of magnitude of the applied magnetic field and effective magnetic field at 2 K.

Comments 9: When coercive field increases with the field tipping angle off the perpendicular direction – the effect is not just the reduction of the perpendicular component of the magnetic field? What is the effect of the field component parallel to the plane of the flake, does it suppresses coercivity? How strong is this effect?

Reply: The effect is that the magnetic moment will tilt with the applied in-plane magnetic field as shown in Fig. 4f, which is the key point in SW model. The effect is clearly described by the standard Stoner-Wohlfarth model (in the single domain regime) and modified Stoner-Wohlfarth model (near the coercivity).

If the field is parallel to the plane of the flake, the condition will be very interesting. At a large field (>6 T), all the magnetic moments are in-plane, therefore the measured R_{xy} (M_z) is around 0. When the magnetic field sweeps from 6 T to 0 T, the magnetic moment will flip equally to both perpendicular directions. Hence R_{xy} (M_z) shows a zero value. From 0 T to -6 T, the R_{xy} (M_z) still keeps zero value for the same reason. This is the measurement results shown in Fig. 4b and Fig. S6d.

This is also the reason why the modified SW model can only work for the fitting of coercivity from 0° to 85° in Fig. 4b.

Comments 10: p. 5 Scaling behavior of FGT switching from 3d to 2d at 7.6nm? more than one point is needed to determine the scaling behavior for thickness below 7.6 nm. The statement “which is not shown in our figure” is ambiguous and therefore confusing. Is the meaning “the data do not show this behavior” or “we do not have the data in this regime”? or something else?

Reply: We thank the reviewer #3 for pointing out this fuzzy description due to our improper explanation.

Revision: We delete the sentence. The revised manuscript provides a more clear explanation. “The fitted $\lambda = 1.66$ is very near the value of a standard 3D Heisenberg ferromagnetism. The good fitting using a single fitting curve also indicates that FGT nanoflakes with a thickness of more than 5 vdW layers are still 3D ferromagnets. If FGT nanoflakes evolve from 3D ferromagnetism to 2D ferromagnetism from 25 nm to 5.8 nm, we should be able to obtain two fitting curves with different critical exponents. However, the data does not show this behaviour, which further confirms 3D magnetism when FGT thickness > 5.8 nm. ”

Comments 11: p. 6 What is the origin of the anisotropy in a flake of FGT? Crystalline or shape/geometric? What is the physical meaning of K_A in the experiment? does it matter? In either case I did not notice this discussion in the manuscript.

Reply: As both bulk and nanoflake FGT show perpendicular anisotropy, its origin is definitely crystalline field. The physical meaning of K_A is the anisotropic energy per volume.

Revision: Based on this comment, the origin of the magnetic anisotropy and the physical meaning K_A have been added in the revised manuscript.

Comments 12: Do the values of the fitting parameters $a(T)$ and $V(T)$ depend on the choice of the coefficient of kT in the modified SW model? If they do, what is their utility? To show the trend with

temperature and angle? Is it important to have a good handle on the size of the first flipping domain?

Reply: Yes, the $a(T)$ and $V(T)$ depend on the choice of the coefficient. $25 k_B T$ corresponds to 100 seconds measuring time, which is the time scale of the real experimental time. We cannot choose $10 k_B T$ or $40 k_B T$, because the corresponding measuring time is unrealistic.

Understanding the dynamics of the magnetic domain is important for the design of the relevant spintronic devices. In a spin orbit torque device, the size of the first flipping domain is related to how large a polarized current is required to flip the magnetic moment of a device.

Comments 13: A question about the model, I hope the authors will clarify it for me. Equation (5) and similar equations in the Supplementary Materials: eq (5) describes the energy difference between states 1 and 2 for a first flipping domain. This domain, when in the state 1, is just a part of the old single domain state. Why its energy should not be just a scaled energy of the single domain state, i.e. $E(\text{domain in state 1}) = K_A V(T) \sin^2(\phi_2 - \theta) - V(T) M_s B \cos(\phi_2)$? Or is it necessary in this case to include the energy of the ferromagnetically ordered sample, as it changes in state 2? Is interaction parameter needed to describe the energy of the first domain in the old single domain state? For the first domain that flips its energy in the excited (barrier) state 2, shouldn't it be $E(2) = K V(T) \sin^2(\phi_2 - \theta) - V(T) M_s B \cos(\phi_2) + a(T) V(T) M_s * M_s * \cos(\phi_1 - \phi_2)$ or something like that? Should there be another similar term in the expression for the small domain in state 1, such as $a(T) V(T) M_s * M_s$?

Reply: In state 1, the first flipping domain aligns with the old single domain state. However, the interaction between the first flipping domain and the surrounding magnetic moments always exists no matter they align with each other or not. We just use a **mean field** method to describe this interaction, which is easier for experimentalists to do fitting and obtain a clear physics picture. The $a(T)B$ here is a combination of the B field and mean field interaction.

Based on the formula $E(2) = K V(T) \sin^2(\phi_2 - \theta) - V(T) M_s B \cos(\phi_2) + a(T) V(T) M_s * M_s * \cos(\phi_1 - \phi_2)$ proposed by reviewer #3, $E(1) = K V(T) \sin^2(\phi_2 - \theta) - V(T) M_s B \cos(\phi_2) + a(T) V(T) M_s * M_s * \cos(\phi_1 - \phi_1)$, the interaction between the first flipping domain and other magnetic moments still exists in state 1, which confirms that Eq.(5) is reasonable.

The formula of $E(2)$ proposed by reviewer #3 is definitely reasonable for modelling. However, it is much more difficult to deal with. The authors only want to get a mean field model to obtain a clear physics picture. More accurate model should be done based on micro-magnetic simulation, which is far beyond the scope of this paper.

Comments 14: Why interaction energy between the first flipping domain and the rest of the sample with old magnetization depends on the applied magnetic field? Also, the angle ϕ_2 is calculated for the single domain case, when magnetization of a whole sample is ϕ_2 away from the field direction. How can we be sure that when a single domain flips, its barrier (highest energy) state will have the same angle as ϕ_2 , for when the whole sample's magnetization rotates? Should the energy be $E(3) = K V(T) \sin^2(\phi_3 - \theta) - V(T) M_s B \cos(\phi_3) + a(T) V(T) M_s * M_s * \cos(\phi_1 - \phi_3)$, where ϕ_3 may not be equal to ϕ_2 ?

Reply: The $a(T)$ does not depend on the applied magnetic field which is clearly shown in Eq.(5). Again, it is a mean field parameter.

The domain dynamics in the system is actually very complex. We should not think that the first flipping domain flips to ϕ_2 , while the magnetic moments of other parts of the FGT flake still points to the ϕ_1 direction. As the magnetic loop of FGT is square shape, all the magnetic moments flip and overcome the barrier (ϕ_2) in a very narrow range of magnetic field. In this situation, using the calculated ϕ_1 and ϕ_2 based on Eq. (4) is a pretty good approximation, which is further confirmed by the very good fittings of the angular depend coercivities from 0° to 85° at various temperatures. To obtain more accurate modelling, miro-magnetic simulation is required, which is far beyond the scope of this paper.

ϕ_3 is not equal to ϕ_2 .

Revision: We add in a paragraph in supplementary materials to explain the approximation of the calculation of ϕ_1 and ϕ_2 . “The domain dynamics in the system is actually very complex. We should not think that the first flipping domain flips to ϕ_2 , while the magnetic moments of other parts of the FGT flake still points to the ϕ_1 direction. As the magnetic loop of FGT is square-shaped, all the magnetic moments flip and overcome the barrier (ϕ_2) in a very narrow range of magnetic field. In this situation, using the calculated ϕ_1 and ϕ_2 based on Eq. S4 is a pretty good approximation.”

Supplementary Material:

Comments 15: fig S5(f) should the symbols be reversed?

Revision: We thank the very careful reading of reviewer #3. We reversed the two symbols in the revised manuscript.

Comments 16: Fig. S5(a) are the 120K shown? It looks like the 150 K data’s “hard phase” has the same coercivity as 100 K data.

Reply: The 120 K loop is actually shown, the 100 K data’s coercivity is a little wider than the 120 K’s. The coercivity of the 150 K data, which has 2 phases, partly overlaps with the 120 K’s.

Comments 17: Fig. S7(b). the figure caption is confusing: (b) Temperature dependence of remanence and R_{xy} curve with 1 T applied magnetic field. Is R_{xy} at 1T labeled as saturated value in the figure? A bit more clarity and consistency between figure and a caption would of value.

Reply: We thank the reviewer for pointing out the inconsistence, we have adjusted the caption and figure to make them more consistent.

Comments 18: The discussion below (at least a condensed and/or selective portion of it, see italicized part) would be commonly placed in Methods and Materials section of the manuscript, instead of supplementary materials:

“Because of the non-symmetry in our nanoflake devices, the measured Hall resistance was mixed with the longitudinal magnetoresistance. We processed the data by using $(R_{xyA}(+B) - R_{xyB}(-B)) / 2$ to eliminate the contribution from the longitudinal magnetoresistance, where R_{xyA} is the half loop sweeping from the positive field to the negative field, R_{xyB} is the half loop sweeping from the

negative field to the positive field, and B is the applied magnetic field. We also measured the R_{xy} (T) at the remanence point for most of the samples. To measure the R_{xy} (T) at the remanence point, the magnetic moment of samples was first saturated by a 1 T magnetic field and then the magnetic field was decreased to zero (the remanence point). Finally, the temperature dependence of the R_{xy} at remanence was measured when the temperature was increased from 2 K to 300 K. In order to eliminate the non-symmetry effect of the device, we measured R_{xy} (remanence) with both 1 T and -1 T saturation. The real R_{xy} (T) at remanence without R_{xx} mixing was calculated using $(R_{xyA}(T) - R_{xyB}(T)) / 2$. Here R_{xyA} and R_{xyB} are the remanence with 1 T and -1 T saturation, respectively.”

How do we know that the procedure described above for measuring remanence as a function of temperature is not effected by hysteretic effects between field and temperature sweeps? Were at least a couple of points checked and consistency established? If so, it would be helpful to put some of the field-sweep remanence points on the “temperature dependent remanence” figures.

Reply: As FGT nanoflake shows no “glassy” behaviour (no relaxation at remanence), the “temperature dependent remanence” measured in our experiments should be equal to the field-sweep remanence points at various temperatures. Based on this comment, we get the field-sweep remanence points from Fig. S6a and Fig. S6b and draw them into the Fig S6e curve, shown in Fig. R9. From the figure we can see that the “temperature dependent remanence” curve meets the discretely measured field-sweep remanence points very well.

During our measurements, the temperature ramping rate is always 3 K/min, this material doesn’t have a temperature dependent hysteretic effect.

Revision: We have placed this part to methods.

Fig. R9 Black dots are remanence points get from Fig. S6a and Fig.S6b. Red curve is the “temperature dependent remanence”, which is the same as Fig. S6e. We can see from the figure the consistency between them is quite well.

References

1. Fei, Z. *et al.* Edge conduction in monolayer WTe_2 . *Nature Phys.* **13**, 677–682 (2017).
2. May, A.F., Calder, S., Cantoni, C., Cao, H. & McGuire, M. A. Magnetic structure and phase stability of the van der Waals bonded ferromagnet $Fe_{3-x}GeTe_2$. *Phys. Rev. B* **93**, 014411 (2016).
3. Yi, J. *et al.* Competing antiferromagnetism in a quasi-2D itinerant ferromagnet: Fe_3GeTe_2 . *2D Materials* **4**, 011005 (2016).

4. Skomski, R., Oepen, H-P. & Kirschner, J. Micromagnetics of ultrathin films with perpendicular magnetic anisotropy. *Phys.Rev.B* **58**, 3223 (1998).
5. O'handley, R. C. *Modern magnetic materials: principles and applications*. (Wiley, New York, 2000).
6. Bertotti, G. *Hysteresis in magnetism: for physicists, materials scientists, and engineers*. (Academic press, New York,1998).
7. León-Brito, N., Bauer, E. D., Ronning, F., Thompson, J. D. & Movshovich, R. Magnetic microstructure and magnetic properties of uniaxial itinerant ferromagnet Fe_3GeTe_2 . *J. Appl. Phys.* **120**, 083903 (2016).

Reviewers' comments:

Reviewer #1 (Remarks to the Author):

The authors have made extensive improvements to the manuscript, including a number of new experimental results, and they have addressed all the concerns I raised in my earlier report. This paper is of great topical interest and I support publication in Nature Communications.

Reviewer #2 (Remarks to the Author):

Authors' have well addressed most of the comments from the reviewers. I have an additional question before recommending for publication. The authors claim that the spin-spin coupling length is 5 layers. However, there is no investigation of the samples with thickness thinner, or even close to 5 layers. I would suggest that the authors need to be a little more conservative here without over analyzing the data and making bold claims. In fact, the analysis on T_c vs thickness is not enough to justify that the samples above 5.8nm is 3D Ising ferromagnet. The authors at least need to do cross checking. For instance, the author could perform a careful temperature dependent measurement near T_c and do a power law fit to extract the critical exponent β , which can tell if FGT nano-flake is 3D ($1/3$) or 2D ($1/8$). I would suggest the author to remove the claim of spin-spin coupling length, improve the discussion on 3D or 2D Ising ferromagnetism, and revise the associate discussion of 2D to 3D crossover as nanoflake thickness which is very speculative anyway.

Reviewer #3 (Remarks to the Author):

I thank the authors for carefully considering the comments from all the referees, and very diligent effort to answer all the comments. I believe that vast majority of comments have been answered satisfactorily. I also changed my original suggestion to submit the manuscript to a more specialized journal. I agree with the authors that in spite of the low temperature of the relevant effects the study of FGT nanoflakes is useful as a demonstration of a system with properties that open a new field of applications. I therefore recommend publishing this paper in Nature Communications.

Reviewers' comments:

Reviewer #2 (Remarks to the Author):

Comments: Authors have well addressed most of the comments from the reviewers. I have an additional question before recommending for publication. The authors claim that the spin-spin coupling length is 5 layers. However, there is no investigation of the samples with thickness thinner, or even close to 5 layers. I would suggest that the authors need to be a little more conservative here without over analyzing the data and making bold claims. In fact, the analysis on T_c vs thickness is not enough to justify that the samples above 5.8nm is 3D Ising ferromagnet. The authors at least need to do cross checking. For instance, the author could perform a careful temperature dependent measurement near T_c and do a power law fit to extract the critical exponent beta, which can tell if FGT nano-flake is 3D (1/3) or 2D (1/8). I would suggest the author to remove the claim of spin-spin coupling length, improve the discussion on 3D or 2D Ising ferromagnetism, and revise the associate discussion of 2D to 3D crossover as nanoflake thickness which is very speculative anyway.

Reply: In response to reviewer #2, we agree it would be highly desirable to provide the standard scaling analysis near T_c to show the transition from 2D to 3D. However, as stated in the paper and for the reasons given below, the authors assert that this analysis is beyond the scope of the current work.

(1) The critical analysis method we used to determine the coupling length is well established and has been discussed/employed in several high level publications^{1,2,3,4}. The coupling length determined using this method is reasonable and agrees with the 10.4 nm thickness R_{xy} vs T curve, which shows a 3D spinwave behaviour. Hence, the method cannot be regarded as merely "speculative".

(2) Reviewer #2 may be unfamiliar with the techniques required in fabricating monolayer devices. The work required to fully address this reviewer point is extensive because -

i. When compared with graphene, Fe_3GeTe_2 (FGT) exhibits significantly stronger interlayer coupling meaning it is extremely challenging to achieve very thin FGT nanoflakes (< 5 nm) by mechanical exfoliation. FGT nanoflakes 7-10 nm in thickness can be routinely obtained and these are perfect for spintronic device fabrication. So far, after experiments of one year, the thinnest flake we have obtained by exfoliation is 5.8 nm.

ii. With the preceding limitation in mind, we are developing plasma etching methods to obtain monolayer, bilayer, and trilayer FGT. However, much work needs to be done to achieve this goal. As iron doesn't have a fluoride with low boiling point, Ar gas must be used

to etch the nanoflakes. Since this can introduce damage, the quality of the etched nanoflake must be confirmed by Raman spectroscopy. Suitable Raman spectra of FGT are unavailable for comparison (because it's a very new 2D material) as are theoretical simulations of the Raman spectra of FGT with different thicknesses. This is a current focus of our work. The thickness of the sample must be known before fabricating devices, so the thickness dependence of Raman spectroscopy of FGT must be obtained. We conservatively estimate that this work will take several months.

iii. After monolayer, bilayer and trilayer FGT nanoflakes have been successfully obtained by plasma etching, processing must be optimised. This includes transfer of the material from etching machine to glove box, hBN encapsulation, and the provision of electric contacts for transport measurements. Due to the complexity of this fabrication sequence, we estimate that realising ultra-thin FGT (< 5.8 nm) devices will take more than six months.

(3) Reviewer #2 may also under-estimate the difficulty and workload of measurements and data analysis.

i. In our transport experiments we measured R_{xy} instead of M . The fitting method using standard scaling analysis (based on M) therefore needs to be modified. This requires additional theoretical work.

ii. Two comprehensive papers^{5, 6} published in 2017 discussed similar scaling analysis on bulk single crystal FGT. To adequately cover the 2D to 3D transition, we need to fabricate 8-10 devices with thicknesses ranging from monolayer to 10 nm then perform 8-10 similar measurement sets. This comprises 8-10 times the workload of the aforementioned papers, amounting to several months work.

(4) The key result of the current paper is that FGT nanoflakes can exhibit hard magnetic properties with square-shaped loop, high coercivity, and strong perpendicular anisotropy. These properties confirm that FGT has great potential for spintronics based on ferromagnetic vdW heterostructures, a new research direction. The transition from 2D to 3D ferromagnetism is a separate important topic from these hard magnetic properties and the applications in spintronics. The comprehensive investigation on the scaling analysis from 2D to 3D transition in FGT nanoflakes should be written in a separate paper, which Lan Wang's group has already been working on for 6 months and may need one more year to finish. It is impossible and not appropriate to present this work in one paragraph in the current paper.

Based on the four points above, we think that the scaling analysis on FGT from monolayer to ~ 10 nm thick devices is far beyond the scope of the current paper. It should be the content of another very important paper.

Revision: Based on the suggestion of reviewer #2, we add a paragraph (on page 6) “Scaling behaviour near the T_C s of samples with thicknesses from monolayer to > 10 nm should reveal the evolution of the magnetism from 3D to 2D with decreasing thickness in FGT. However, due to relatively strong interlayer coupling in FGT nanoflakes, it is very difficult to obtain FGT nanoflakes with a thickness of smaller than 5 nm by mechanical exfoliation. The focus of this paper is revealing the hard magnetic properties of FGT nanoflakes and their suitability for future spintronic applications. Hence, we propose this scaling analysis as future work.”

References

1. Zhang, R. & Willis, R. F. Thickness-dependent Curie temperatures of ultrathin magnetic films: effect of the range of spin-spin interactions. *Phys. Rev. Lett.* **86**, 2665 (2001).
2. Almahmoud, E., Kornev, I. & Bellaiche, L. Dependence of Curie temperature on the thickness of an ultrathin ferroelectric film. *Phys. Rev. B* **81**, 064105 (2010).
3. Yin, L. *et al.* Magnetocrystalline anisotropy in permalloy revisited. *Phys. Rev. Lett.* **97**, 067203 (2006).
4. Weschke, E. *et al.* Finite-size effect on magnetic ordering temperatures in long-period antiferromagnets: Holmium thin films. *Phys. Rev. Lett.* **93**, 157204 (2004).
5. Liu, B. *et al.* Critical behavior of the van der Waals bonded high T_C ferromagnet Fe_3GeTe_2 . *Sci. Rep.* **7**, 6184 (2017).
6. Liu, Y., Ivanovski, V. & Petrovic, C. Critical behavior of the van der Waals bonded ferromagnet $Fe_{3-x}GeTe_2$. *Phys. Rev. B* **96**, 144429 (2017).

Reviewers' Comments:

Reviewer #2 (Remarks to the Author):

The authors may have misunderstood my comments. I did not ask the authors to add additional results of flakes thinner than 5 nm, which I understand is beyond the scope of the work. Rather, I simply asked the authors to be cautious about their claim and turn down the tone. Although the response is not satisfactory (since the authors misunderstood my comments), the work is interesting enough for Nature Communication and I recommend it for publication. I still hope the authors could make some minor revision to be precise about what they can claim. For instance, to remove the claim of spin-spin coupling length in the abstract. The author writes “By employing criticality analysis, the existence of magnetic coupling with a coupling length of ~ 5 vdW layers between vdW atomic layers is confirmed in FGT.” I think it is not justified to use the word “confirmed” since there is no experiment data of thickness below 5 nm for confirmation. I suggest replacing “confirmed” with “estimated”.